# Modulation of peroxisomal import by the PEX13 SH3 domain and a proximal FxxxF binding motif

Stefan Gaussmann [1,2,5], Rebecca Peschel [3,5], Julia Ott[3], Krzysztof M. Zak[2], Judit Sastre[4], Florent Delhommel [1,2], Grzegorz M. Popowicz [1,2], Job Boekhoven [4], Wolfgang Schliebs[3], Ralf Erdmann [3] ✉ & Michael Sattler [1,2] ✉

Import of proteins into peroxisomes depends on PEX5, PEX13 and PEX14. By combining biochemical methods and structural biology, we show that the C-terminal SH3 domain of PEX13 mediates intramolecular interactions with a proximal FxxxF motif. The SH3 domain also binds WxxxF peptide motifs in the import receptor PEX5, demonstrating evolutionary conservation of such interactions from yeast to human. Strikingly, intramolecular interaction of the PEX13 FxxxF motif regulates binding of PEX5 WxxxF/Y motifs to the PEX13 SH3 domain. Crystal structures reveal how FxxxF and WxxxF/Y motifs are recognized by a non-canonical surface on the SH3 domain. The PEX13 FxxxF motif also mediates binding to PEX14. Surprisingly, the potential PxxP binding surface of the SH3 domain does not recognize PEX14 PxxP motifs, distinct from its yeast ortholog. Our data show that the dynamic network of PEX13 interactions with PEX5 and PEX14, mediated by diaromatic peptide motifs, modulates peroxisomal matrix import.

Peroxisomes are single membrane-enveloped organelles of eukaryotic cells essential for several metabolic pathways, mainly related to lipid metabolism and the removal of toxic oxidation products[1–3]. The physiological importance of these highly conserved organelles is emphasized by diseases such as Zellweger Spectrum Disorders (ZSD) that result from defects in peroxisome biogenesis[4]. Biogenesis and function of peroxisomes rely on peroxisome-related proteins called peroxins[5] that are involved in membrane assembly and post-translational matrix protein import into the organelle[6]. Human and yeast peroxins are abbreviated as "PEX" and "Pex", respectively. PEX13, a peroxin crucial for peroxisomal import, has been linked to Zellweger spectrum disorder[7] and malfunction of PEX13 leads to impaired

biogenesis and neonatal death[8]. The general mechanisms of peroxisomal biogenesis and matrix protein import are evolutionarily conserved. Peroxisomal matrix localization of enzymes depends on conserved C- (PTS1) or N-terminal (PTS2) peroxisomal targeting signals[9,10]. Cytosolic PTS1-cargo proteins are recognized by the C-terminal tetratricopeptide repeat (TPR) domain of the peroxisomal receptor PEX5[11]. The receptor-cargo complex is tethered to the peroxisomal membrane via the intrinsically disordered PEX5 N-terminal domain (NTD)[12] (Fig. 1a). At the peroxisomal membrane, PEX5 NTD interacts with the membrane-bound components of the translocon, PEX14 and PEX13[13–19] (Fig. 1a). These interactions may be regulated by weak interactions of PEX5 and PEX14 with peroxisomal membranes[20].

[1]Technical University of Munich, TUM School of Natural Sciences, Bavarian NMR Center and Department of Bioscience, Lichtenbergstr. 4, 85747 Garching, Germany. [2]Helmholtz Munich, Molecular Targets and Therapeutics Center, Institute of Structural Biology, Ingolstädter Landstr. 1, 85764 Neuherberg, Germany. [3]Institute of Biochemistry and Pathobiochemistry, Department of Systems Biology, Faculty of Medicine, Ruhr University Bochum, 44780 Bochum, Germany. [4]Technical University of Munich, TUM School of Natural Sciences, Department of Chemistry, Lichtenbergstr. 4, 85747 Garching, Germany. [5]These authors contributed equally: Stefan Gaussmann, Rebecca Peschel. ✉e-mail: ralf.erdmann@rub.de; sattler@helmholtz-munich.de

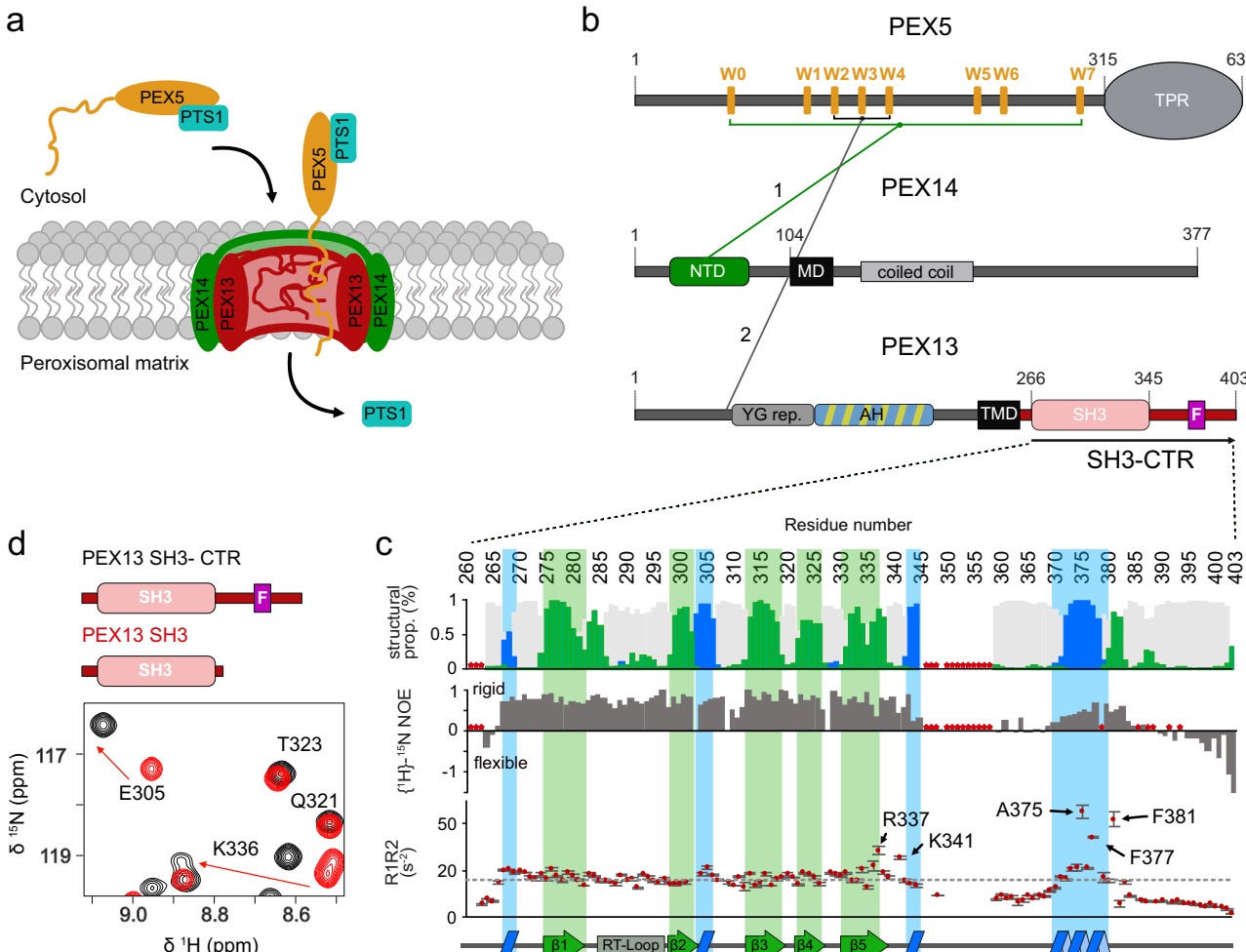

**Fig. 1 | Schematic representation of PTS1 import with interactions of the core components and NMR-based analysis of the conformation and dynamics of PEX13 C-terminal region. a** Schematic overview of cargo recognition, docking and cargo translocation. **b** Schematic representation of domain architecture and intermolecular interactions of the human peroxins PEX5, PEX14 and PEX13 respectively. Lines 1 and 2 between the peroxins indicate known binding events involving the targeted structure elements or motifs[12,14–16,20]. **c** Asterisks indicate proline or missing assignment. Top: $^{13}$C secondary chemical shifts ($\Delta\delta^{13}C_\alpha - \Delta\delta^{13}C_\beta$) analysed with TALOS + . The propensity for the secondary structure elements random coil, α-helix or β-sheet are represented in gray, blue or green, respectively. Our data support the typical β-sandwich fold of the SH3 domain and the presence of a short α-helical motif comprising the FxxxF motif. Middle: elevated {$^1$H}-$^{15}$N heteronuclear NOE values indicate an extended SH3 fold (265-345) and a folded FxxxF motif with similar values to the SH3 domain. Bottom: $^{15}$N $R_1*R_2$ relaxation rates as a function of amino acid sequence. SH3 core residues (266-335) show an average of 16.6. C-terminal residues R337, K341, A375, F377 and F381 show values of $28.4 \pm 1.8$, $25.2 \pm 0.6$, $45 \pm 2.6$, $34,2 \pm 0.6$ and $42.0 \pm 3.0$, respectively. Error bars are calculated from fitting errors of $R_1$ and $R_2$ values. Values higher than the average in structured regions indicate the presence of conformational dynamics/and or transient interactions. Secondary structure elements are illustrated at the bottom. **d** Zoomed view of NMR spectra of PEX13 SH3-CTR (black) and PEX13 SH3 (red). Source data are provided as a Source Data file.

It was recently proposed that the PEX5 cargo complex translocates into the peroxisomal matrix via a biomolecular condensate phase, which involves YG-repeats in PEX13[21–23] (Fig. 1a).

Many aspects of peroxisome biogenesis have been studied in yeast, where Pex5/cargo docking is mediated by a membrane-associated complex consistent of the Pex13 SH3, the Pex14 NTD and Pex5 NTD, essential for both PTS1 and PTS2 import[17–19,24]. The interactions within the docking complex are mediated by (di)aromatic penta peptide motifs ("WxxxF/Y") of Pex5 with the N-terminal domain of Pex14 and a poly-proline (PxxP) motif of Pex14 that binds to the Pex13 SH3 domain[25,26]. However, the minimal pore reconstituted in vitro comprises only Pex5 and Pex14[27]. The yeast docking complex is distinct from the human one by the additional presence of Pex17, which is essential for the peroxisomal import in yeast[28], but not found in humans[29]. The conservation of homologous interactions in humans is not well understood. Interactions of PEX5 (di)aromatic motifs and the globular N-terminal domain of PEX14 (Fig. 1b, 1) are conserved, but

a docking complex with PEX13, similar to the one observed in yeast, has not been reported. Human PEX13 is an integral membrane protein with an intrinsically disordered N-terminal region harboring several YG repeats followed by an amphipathic helix (AH), a transmembrane region[22,23], and a mostly unstructured C-terminus with an FxxxF motif and a SH3 domain. (Fig. 1b). Such an FxxxF motif is not found in yeast or invertebrates such as insects. Recent studies have demonstrated that yeast Pex13 can mediate liquid-liquid phase separation with a potential role to peroxisomal import. However, the conservation of these mechanisms and interactions of human PEX13 with other peroxins are currently poorly understood. An early study postulated that a PEX5/PEX13 interaction involves a subset of the eight PEX5 WxxxF/Y motifs, namely W2, W3 and W4, with an N-terminal region in PEX13 (Figs. 1b, 2), which was shown to be essential for catalase import[12]. Furthermore, a Zellweger mutation inducing a W313G substitution in the PEX13 SH3 domain did not abolish the interaction with PEX14 and was demonstrated to disrupt PTS1 but not PTS2 import[30,31].

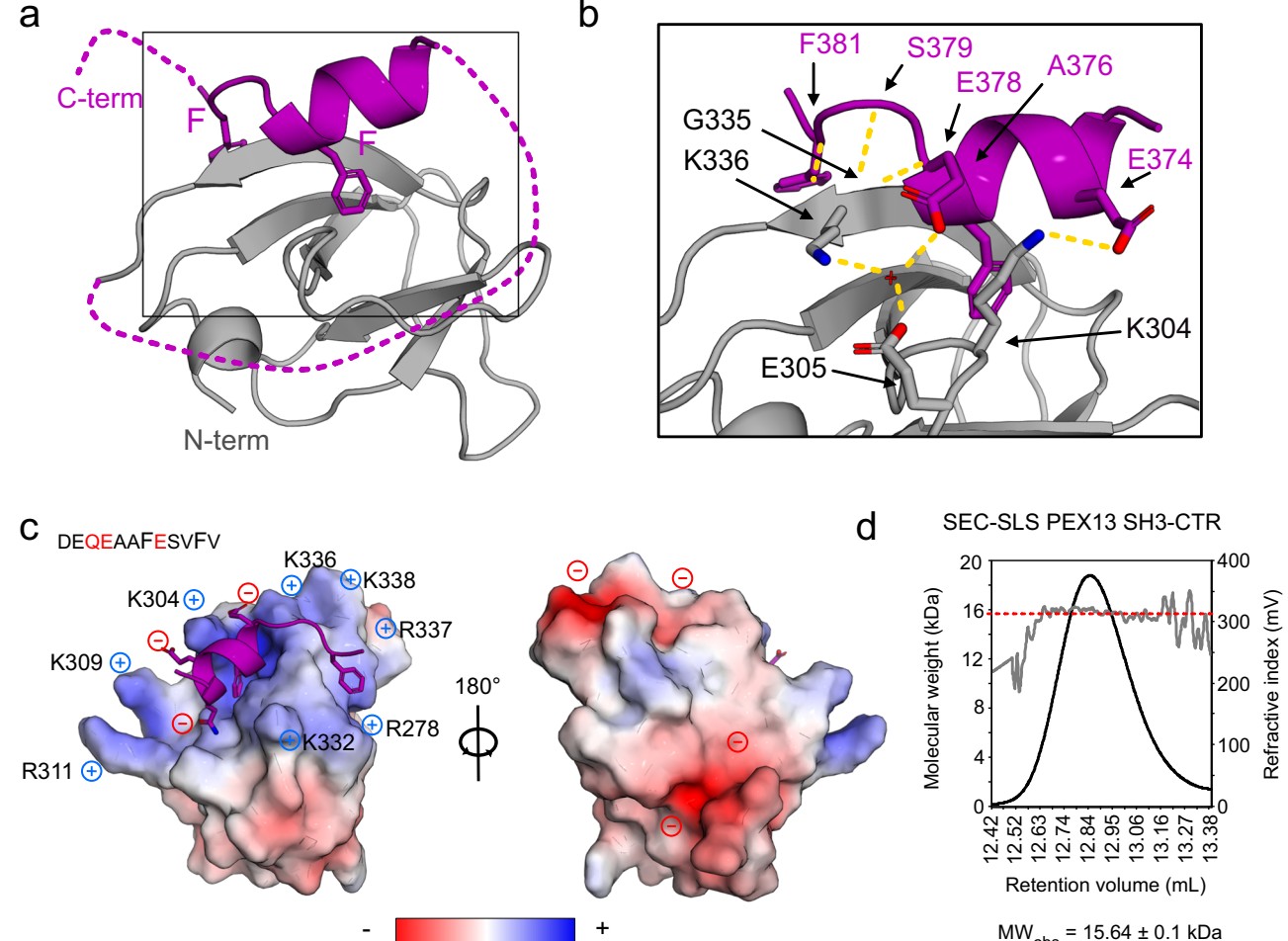

**Fig. 2 | Structural analysis of PEX13 SH3 in complex with FxxxF motif.**
**a** Crystallographic structure of PEX13 SH3 2GS FxxxF showing the α-helical FxxxF motif, which clamps β1 and β5 between the two Phe residues. **b** Zoom view visualizing the hydrogen bond network between the SH3 domain and the FxxxF peptide. Polar backbone contacts are mediated by A376 and G335 as well as S379, F381 and K336 and sidechains E374 and E378 are coordinated by K304, E305, K336 and a water molecule. **c** Electrostatic surface representation showing the positively

charged FxxxF binding site which is caused by seven Arg or Lys residues. The peptide in contrast, is negatively charged which is favored for the binding (Q373, E374 and E387). A 180° rotation on the Y axis reveals a negatively charged backside. **d** Static light scattering analysis of PEX13 SH3-CTR shows the molecular weight (red dashed line) of 15.64 ± 0.1 kDa versus the calculated mass of 15.56 kDa indicating a monomeric state. Source data are provided as a Source Data file.

In this study, we present a high-resolution structure of the human PEX13 SH3 domain with an intramolecular FxxxF motif, demonstrating the structural basis of this non-canonical interaction with an SH3 domain. We identify molecular interactions of PEX13 with PEX5 and PEX14 by combining biophysical methods, structural biology, and functional analysis in cells. We show that the FxxxF peptide motif in a C-terminal extension of PEX13 modulates the binding of WxxxF/Y motifs to the PEX13 SH3 domain. Our binding studies demonstrate that a dynamic network of interactions of the PEX5 NTD with PEX13 and PEX14 plays important roles in peroxisomal protein import.

## Results

### The PEX13 C-terminal region harbors an SH3 domain followed by a FxxxF motif

To study the conformation of the PEX13 C-terminal region, comprising a SH3 domain and a C-terminal unstructured extension (SH3-CTR), we used NMR spectroscopy. The $^1$H-$^{15}$N correlation spectra of the PEX13 SH3-CTR (residues 261-403) show well dispersed signals and additional signals with narrow linewidth, which correspond to the globular SH3 domain and the disordered C-terminal region, respectively (Supplementary Fig. 1a). NMR $^{13}$C$_\alpha$ and $^{13}$C$_\beta$ secondary chemical shifts are consistent with the secondary structure of SH3 domains composed of

five β-strands (β1 to β5)[32]. An additional α-helical region located in the C-terminal region (371-383) downstream of the SH3 domain encompasses an FxxxF motif (Fig. 1c, Top; Supplementary Fig. 1c), which is highly conserved across mammals (Supplementary Fig. 1c), but not found in yeast or invertebrates. Upon comparison of NMR spectra of the PEX13 SH3-CTR and the SH3 domain only, significant chemical shift differences are seen, which map to the strands β1, β5 and the region of β2 and a flanking turn. (Fig. 1d; Supplementary Fig. 1a, d). NMR {$^1$H}-$^{15}$N heteronuclear NOE (hetNOE) data, which report on backbone dynamics at sub nanosecond timescales, indicate that the PEX13 SH3 fold extends beyond the typical SH3 structural elements compared to the yeast Pex13 and other human SH3 domains (Supplementary Fig. 1e). In the C-terminal region, around residue 360, low hetNOE values indicate high conformational flexibility of the C-terminal region. However, the FxxxF motif in this region shows large hetNOE values of ~0.8, comparable to those seen in the globular SH3 domain (Fig. 1c, middle), demonstrating that this region is rigid. This is further supported by the product of NMR $^{15}$N $R_1$ and $R_2$ relaxation rates, where average values around 16.6 s$^{-2}$ are seen for the rigid core domain. In contrast, some C-terminal residues of the SH3 domain and the FxxxF motif show elevated values ranging from 25 s$^{-2}$ to 45 s$^{-2}$, indicative for the presence of conformational dynamics (Fig. 1c, bottom)[33].

Taken together these data suggest a potential intramolecular interaction between the FxxxF motif and the PEX13 SH3 domain.

## The PEX13 FxxxF motif forms an intramolecular interaction with the SH3 domain

The domain boundaries defined by the NMR analysis were used to generate chimeric constructs which contain the PEX13 SH3 domain and the FxxxF motif ($D_{371}$EQEAAFESVFV$_{383}$) separated by GGGGS (GS) linkers. Structures of the apo SH3 domain (Supplementary Fig. 2a) and SH3-(GS)$_2$-FxxxF (Fig. 2a) were solved by X-ray crystallography at 1.8Å and 2.3Å resolution, respectively (Supplementary Table 1). A comparison of the apo PEX13 SH3 and the complex structure show that the SH3 fold is highly similar (backbone coordinate RMSD = 0.39) (Supplementary Fig. 2c). Analysis of the structure of the SH3 domain shows a network of polar contacts between the N- and C-terminal regions stabilizing the β1/β5 interaction (Supplementary Fig. 2b), consistent with the extended domain boundaries observed by NMR. The structure of the complex shows interactions of the α-helical FxxxF motif and SH3 domain mediated by hydrophobic contacts of the two phenylalanines, which clamp around β1 and β5 (Fig. 2a, c), and polar interactions involving sidechain and backbone contacts. Backbone hydrogen-bonds are formed between A376 and G335 as well as S379, F381 and K336. The negatively charged sidechains E374 and E378 show electrostatic interactions with K304, E305, K336, and a water molecule (Fig. 2b). Interestingly, nine out of the eleven Arg and Lys residues are located at the FxxxF binding surface, forming a positively charged surface area, which is favorable for binding negatively charged peptides such as the C-terminal FxxxF motif (Fig. 2c). In contrast, the PxxP binding site located at the opposite side of the SH3 domain, is mostly negatively charged. The crystal structure was confirmed in solution by NMR titration experiments of the isolated SH3 domain with an FxxxF peptide (350-403). Strong chemical shift perturbations map to the binding site seen in the crystal structure, and spectral changes at saturated binding are very similar to the native SH3-CTR protein (Supplementary Fig. 2d, e). Static light scattering (SLS) of PEX13 SH3-CTR indicates a molecular weight of $15.6 \pm 0.1$ kDa, which correlates well with the calculated mass of 15.6 kDa (Fig. 2d). This confirms that the FxxxF/SH3 interaction occurs intramolecularly and does not involve oligomerization of the construct at the measured concentration. These results suggest that the PEX13 SH3-CTR adopts a closed state, with an intramolecular interaction of the C-terminal FxxxF motif with the SH3 domain. Notably, the human PEX13 SH3 interacts with (di)aromatic peptide motifs on a surface opposite to the PxxP binding region, as has previously been reported for yeast Pex13[26].

## The PEX13 FxxxF motif binds to the PEX14 NTD

We next evaluated possible interactions of PEX13 with other components of the import machinery. First, we analyzed the PEX13 SH3-CTR / PEX14 NTD interaction by NMR titrations monitoring effects on the $^{15}$N labeled PEX14 NTD upon titration of unlabeled PEX13 SH3-CTR. Significant chemical shift perturbations are observed (Fig. 3a; Supplementary Fig. 3a), very similar to the those seen for the interaction of PEX14 NTD with the PEX19 FxxxF motif (Fig. 3c)[15]. This is consistent with highly similar amino acid sequences of the two motifs (Fig. 3b) with four identical and two similar residues (Fig. 3b). Not surprisingly, mapping of the chemical shift perturbations onto the PEX14 NTD structure highlights the involvement of key residues that are also involved in binding of PEX5 WxxxF/Y motifs[14,15]. The binding affinity for the PEX14 NTD / PEX13 FxxxF interaction was determined by ITC. The PEX13 SH3-CTR (261-403) or a PEX13 FxxxF peptide (residues 350-403) bind to PEX14 NTD with a $K_D$ of 5.4 μM and 2.8 μM respectively (Fig. 3e; Supplementary Table 2).

Although the dissociation constants ($K_D$) are in a similar range, the thermodynamic features are notably different. While the interaction with PEX13 SH3-CTR profits from enthalpic and entropic effects, which

likely reflects the transition of the PEX13-associated FxxxF motif to the PEX14 bound form, the interaction with free FxxxF peptide comes with an entropic penalty, due to a free-to-bound transition of the FxxxF motif (Fig. 3f; Supplementary Table 2). The ITC experiments further demonstrate a stoichiometry of 1:1 in both cases (Fig. 3e lower panel). Interestingly, the binding affinity of the PEX14 NTD towards the PEX13 FxxxF motif is 3 times stronger compared to the affinity with the PEX19 FxxxF motif ($K_D = 9.2$ μM, Neufeld, et al. [15]).

To assess whether the interaction site in PEX13 is limited to the FxxxF motif or involves additional regions, we titrated unlabeled PEX14 NTD (1-104) onto $^{15}$N labeled PEX13 SH3-CTR (Fig. 4). We observed strong chemical shift perturbations not only for the FxxxF motif but also for residues in the SH3 domain (Fig. 4a, b; Supplementary Fig. 4a). Notably, NMR signals of the SH3 domain shift towards their position seen in the isolated SH3 domain, indicating that the SH3 domain does not contribute to the PEX14 NTD interaction (Fig. 4b, c yellow boxes). NMR signals of residues in the FxxxF motif (370-386) experience large chemical shift perturbations or line-broadening for residues 378 to 383 located in the core motif (Fig. 4c). From these data we conclude that the PEX14 NTD interacts solely with the FxxxF motif, and that the intramolecular interaction of the FxxxF motif in the closed structure of the PEX13 SH3-CTR is released upon PEX14 NTD binding. As a result, the binding surface of the PEX13 SH3 domain is available to interact with other (di)aromatic peptide ligands. (Fig. 4d).

## PEX13 SH3 interactions with poly-proline motifs

In yeast, the Pex14/Pex13 interaction is mediated by a class II poly-proline motif of Pex14, which binds to the Pex13 SH3 domain[26]. Human PEX14 also harbors a PxxP motif downstream of its NTD (residues 87-102), similar to that in yeast (residues 85-94). NMR titrations with unlabeled PEX14 NTD$_{long}$ (1-113) onto $^{15}$N labeled PEX13 SH3 were used to evaluate this potential interaction. No significant chemical shift perturbations were observed, showing that this interaction is not conserved from yeast to human (Supplementary Fig. 4b). Our results are in agreement with previous experiments using co-immunoprecipitation that showed an interaction of PEX14 with the PEX13 Zellweger mutant W313G. This mutation is located in the SH3 domain and destabilizes the SH3 fold. Interestingly, the mutation does not affect the human PEX14/PEX13 interaction, while the corresponding mutation in yeast abolished Pex14 binding[31]. We thus investigated whether human PEX13 SH3 has any ability to bind PxxP motifs. Interestingly, PxxP motifs present in the N-terminal region of PEX13 show significant binding in NMR titration experiments with the PEX13 SH3 domain and the PEX13 SH3-CTR. These experiments demonstrate that, in principle, the human SH3 domain can mediate PxxP interactions and that these are independent, and non-overlapping, with the FxxxF binding (Supplementary Fig. 4c–e). However, chemical shift perturbations mapped onto a structural model of a class II PEX13 SH3/PxxP complex do not well agree in the second proline binding pocket, which is occupied with an isoleucine residue. This may suggest that the human PEX13 SH3 domain prefers binding to class I PxxP motifs, distinct from the yeast orthologue (Supplementary Fig. 4f).

## PEX5 WxxxF/Y motifs compete with the PEX13 FxxxF motif on PEX13 SH3

We then characterized the molecular interactions between PEX13 and PEX5. Using NMR titrations, the PEX5 NTD (1-315) but not PEX5 TPR (315-639) domain was found to bind to PEX13 SH3-CTR and PEX13 SH3 (Supplementary Fig. 5). We show that all eight WxxxF/Y motifs of PEX5, also known as W-motifs, bind to the PEX13 SH3 domain or the PEX13 SH3-CTR. Notably, the PEX5 WxxxF and FxxxF binding sites overlap and are thus competitive (Fig. 2a; Fig. 5; Supplementary Fig. 6). The PEX5 W4 motif was identified as the strongest binder followed by W2 and W3 (Fig. 5b). These PEX5 motifs can compete with the PEX13 C-terminal FxxxF motif, while other (di)aromatic

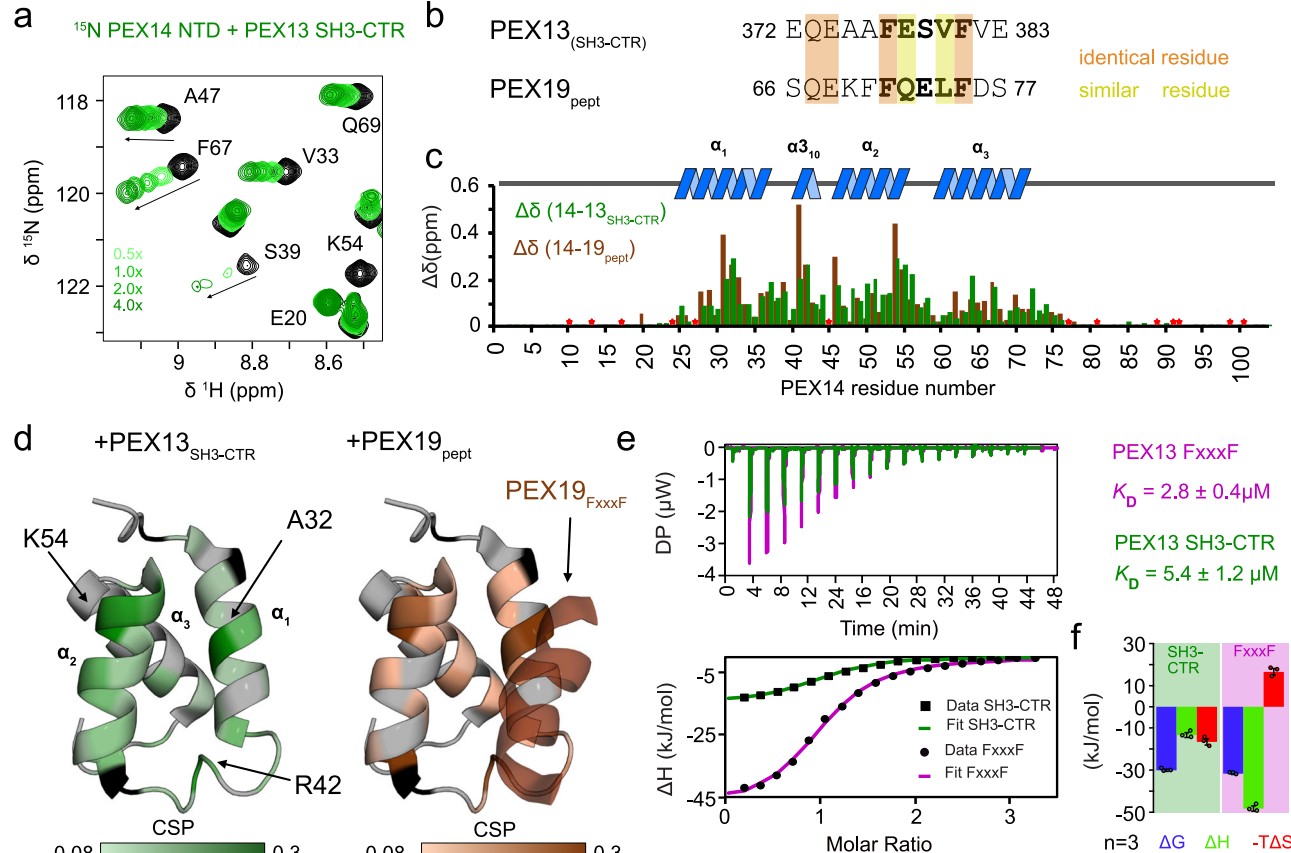

**Fig. 3 | Interaction of PEX14 NTD with PEX13 SH3-CTR in comparison with PEX19 (66-77). a** $^1$H,$^{15}$N correlation spectra of $^{15}$N-labeled recombinant PEX14(NTD) free (black), and in complex with PEX13 SH3-CTR (green scale). **b** Sequence alignment of PEX13 and PEX19 FxxxF motifs. Red and yellow boxes indicate identical and similar residues. **c** NMR chemical shift perturbations of PEX14 NTD in the presence of PEX13 SH3-CTR (green) or PEX19 peptide (brown) (Neufeld et al.[15]) plotted on the sequence with indicated secondary structure elements above. Asterisk indicate proline or missing assignment. **d** Chemical shift perturbations (0.08 to 0.3 ppm) of

PEX13 SH3-CTR (left) and PEX19 peptide (right) mapped on the PEX14 NTD/PEX19 66-77 structure (2w85). **e** ITC experiments of PEX14 NTD with PEX13 SH3-CTR (green) and PEX13 FxxxF (pink) showing very different energetics but the same one to one stoichiometry. **f** Energetic contribution of the PEX14 NTD interaction with PEX13 SH3-CTR (left graph) and PEX13 FxxxF peptide (right graph). Average values and error bars (SD) are calculated from 3 different experiments ($n$ = 3). Source data are provided as a Source Data file.

peptide motifs only bind to PEX13 SH3 with low affinity and not to PEX13 SH3-CTR (Fig. 5b). PEX5 W3 induces small CSPs but significant line-broadening on PEX13 SH3 (Fig. 5b star), indicating strong binding as well. These observations are supported by ITC experiments, which show binding affinities with $K_D$ = 43, 88, and 102 µM for W4, W2 and W3, respectively. To relate these affinities, we evaluated the binding of the PEX13 FxxxF motif in *trans*, which shows a $K_D$ = 27 µM (Fig. 5d; Supplementary Table 3; Supplementary Fig. 6d), and thus stronger than any of the (di)aromatic peptide motifs of PEX5. Affinities of the other motifs were too weak to be measured by ITC (Fig. 5d; Supplementary Fig. 6c). Binding of the same PEX5 motifs to PEX14 NTD is overall much stronger, ranging from 21 to 3136 nM (Fig. 5d, e; Supplementary Fig. 7; Supplementary Table 4). These values are in agreement with previous studies in different buffer conditions[34]. Of note, amongst the eight motifs in PEX5, W4 shows the highest relative binding affinity for PEX13 SH3 and the weakest interaction with the PEX14 NTD (Fig. 5d). To investigate the structural basis for the PEX5 W4 / PEX13 SH3 interaction, we determined the crystal structure of a PEX13 SH3 GS W4 chimera (Fig. 5f; Supplementary Table 1). In contrast to the PEX13 FxxxF motif, the binding interface is limited to the core motif driven by hydrophobic interactions and few hydrogen bonds from R183, Y185, and Y188 to the backbone or K304 sidechain (Fig. 5g). Nevertheless, polar backbone interactions involving G335 and K336 and coordination from

K304 seem important since they are conserved from PEX13 FxxxF to PEX5 W4 (Fig. 5h, colored lines). These results show that all (di)aromatic peptide motifs in the PEX5 NTD can bind the isolated PEX13 SH3 domain, but only W4, W2, and W3 can compete with the intermolecular PEX13 FxxxF motif. The structure of the PEX13 SH3 -W4 complex shows a limited binding interface that lacks the electrostatic interactions that are seen with the PEX13 FxxxF motif (Fig. 2a).

**The PEX13 FxxxF motif modulates PTS1 import**

The functional significance of the PEX13 SH3-CTR and the intramolecular interactions of the proximal FxxxF with the SH3 domain motif were addressed using a cellular protein import complementation assay (Figs. 6, 7). Here, the complementing activity of PEX13 variants was analyzed using T-REx 293 PEX13 KO cells, which are characterized by non-functional peroxisomal PTS1 import[35]. These PEX13-deficient cells were transfected with a bicistronic vector coding for (i) GFP-SKL as a marker for peroxisomal PTS1 import and (ii) full-length PEX13 (PEX13 FL) or PEX13 truncation and mutation variants (Fig. 7d). The import efficiency of cells expressing different PEX13 variants was monitored by fluorescence microscopy and evaluated with the Pearson Colocalization Coefficient (PCC) using automated correlation of the PMP70 (Alexa Fluor 594) and GFP fluorescence signals. The PCC ranges from 1=perfect colocalization over 0=random distribution to −1=no colocalization (Fig. 7c).

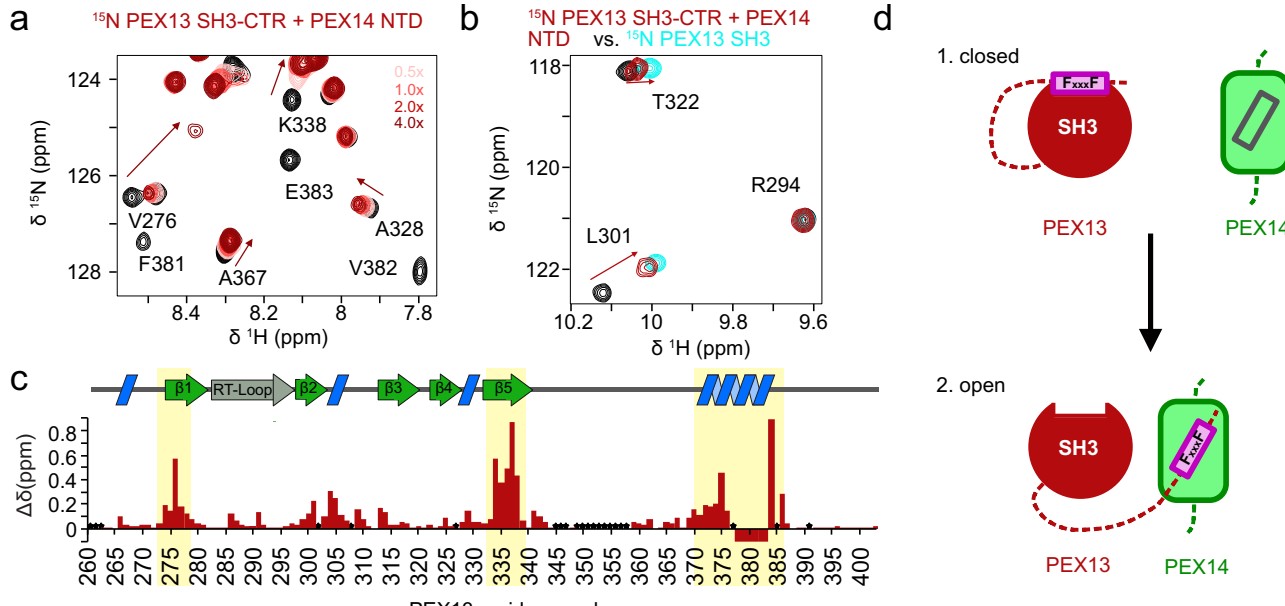

**Fig. 4 | NMR titration of PEX14 NTD onto 15N PEX13 SH3-CTR. a** Spectra overlay from NMR titration of unlabeled PEX14 NTD onto $^{15}$N labeled PEX13 SH3-CTR where large chemical shift perturbations of resonances from the FxxxF motif were observed. **b** Spectra overlay of free PEX13 SH3-CTR (black), PEX13 SH3-CTR + 4x PEX14 NTD (dark red) and apo PEX13 SH3 (blue) showing the transfer from the closed conformation of the PEX13 SH3-CTR back to apo form of PEX13 SH3.

**c** Chemical shift perturbations mapped on the sequence and structural elements (above) of PEX13 SH3-CTR visualizing the effect of the opening on the structural elements β1, β5 and the FxxxF motif. Signals that experience large chemical shift perturbations or line-broadening down to beyond detection are indicated with negative values. **d** Schematic representation of the opening process. Source data are provided as a Source Data file.

Cells solely expressing GFP-SKL (-), which served as negative control, showed no import activity, indicated by the diffuse distribution of GFP in the cytosol, while the punctuate pattern of the peroxisomal membrane marker PMP70 indicates the presence and location of peroxisomal membrane ghosts (Fig. 6a). In contrast, transfection with PEX13 full length, which was used as positive control, resulted in the reconstitution of peroxisomal import in most cells, represented by the congruent punctate pattern of GFP and colocalization with the peroxisomal marker protein PMP70 (Fig. 6b).

To assess the importance of the PEX13 C-terminal region including the CTR or solely the SH3 domain, the complementation activity of truncation variants or mutants of PEX13 was assessed. Cells expressing PEX13 ΔSH3-CTR, lacking the SH3-domain as well as the C-terminal extension showed a significantly reduced capability for the import of the peroxisomal marker protein GFP-SKL into peroxisomes ($p < 0.05$) compared to the full-length PEX13 (Figs. 6c, 7c). A similar significant reduction of GFP-SKL import ($p < 0.01$) was also observed for PEX13 ΔSH3, which lacks the SH3 domain (Fig. 6d, Fig. 7c). However, transfection of PEX13 KO cells with the PEX13 ΔCTR, containing the SH3 domain but lacking the C-terminal region with the FxxxF motif, showed a median PPC similar to PEX13 FL with a higher population of cells with reduced import efficiency (Supplementary Fig. 8a; Fig. 7c).

To investigate the role of the FxxxF/WxxxF/Y motif in the PEX13 CTR, mutants PEX13 FF/W4 and PEX13 IA/CC were analyzed (Fig. 7c). In PEX13 FF/W4, the FxxxF core motif (FESVF) was replaced with W4 (WYDEY), which enhances the affinity to the SH3 domain 4.5-fold ($K_D = 27\,\mu M$ for FF vs. $6\,\mu M$ for FF/W4) (Supplementary Table 3, Supplementary Fig. 9a, Supplementary Fig. 6d). The double cysteine mutant PEX13 IA/CC introduces two cysteines (I333C/A376C), which form a disulfide bridge in oxidizing conditions. This locks the FxxxF motif onto the SH3 domain and thereby inhibits PEX5 binding. Under reductive conditions, the lock opens and allows PEX5 to bind (Supplementary Fig. 9b). Both mutants showed a slight, but not significant reduction of peroxisomal import of GFP-SKL (Fig. 7a, c; Supplementary Fig. 8b).

Next, we investigated the functional effects of (i) inactivation of the C-terminal FxxxF motif, which would abolish its intramolecular interaction with the SH3-domain but also intermolecular interactions with the PEX14 NTD, by replacing the PEX13 FxxxF motif by polyA (FF/A5) (Supplementary Fig. 9c), and (ii) mutation of the corresponding binding site on the SH3-domain, which impair the intramolecular FxxxF motif interaction but also binding to other (di)aromatic penta peptides in PEX5. For this a PEX13 (YEG) triple mutant was created, which lost the ability to bind the PEX13 FxxxF motif and the PEX5 W4 motif but not a PEX13 PxxP motif (Supplementary Fig. 9d–f). In addition, we tested a PEX13 YEG-FF/A5 mutation, as loss of function affecting both the FxxxF motif and the corresponding binding site in the SH3-domain. Interestingly, both mutants and their combination showed a slight but statistically not significant import defect (Fig. 7b, c; Supplementary Fig. 8c, d).

As outlined above, the automated analysis of fluorescence microscopy images indicated that several mutations influenced the functional performance of PEX13 in peroxisomal protein import. To further assess the influence of our mutations on peroxisomal matrix import efficacy, we performed digitonin fractionation of cells with subsequent analysis via immunoblotting to dissect the localization of GFP. As positive and negative control served unmodified T-REx 293 cells endogenously expressing PEX13 wild type (WT) and PEX13 KO cells transfected with solely GFP-SKL (-), respectively (Fig. 8a). For the positive control (FL), we observed GFP almost exclusively in the organellar (O) fraction. Analysis of the negative control (-) showed all GFP in the cytosolic fraction (C) (Fig. 8a, b). The samples of PEX13 FL expressing PEX13KO cells show about 2x more organellar GFP than cytosolic GFP (Fig. 8a, b). This indicates that the import defect of the mutant was complemented, however, the complementation was not 100%. This observation agrees with the fluorescence microscopy images (Fig. 6b), which showed a punctate staining for most cells, but some exhibited cytosolic mislocalization of the peroxisomal marker. Drastic effects on the efficacy of peroxisomal import of GFP-SKL were observed in cells expressing the mutants ΔSH3-CTR and ΔSH3

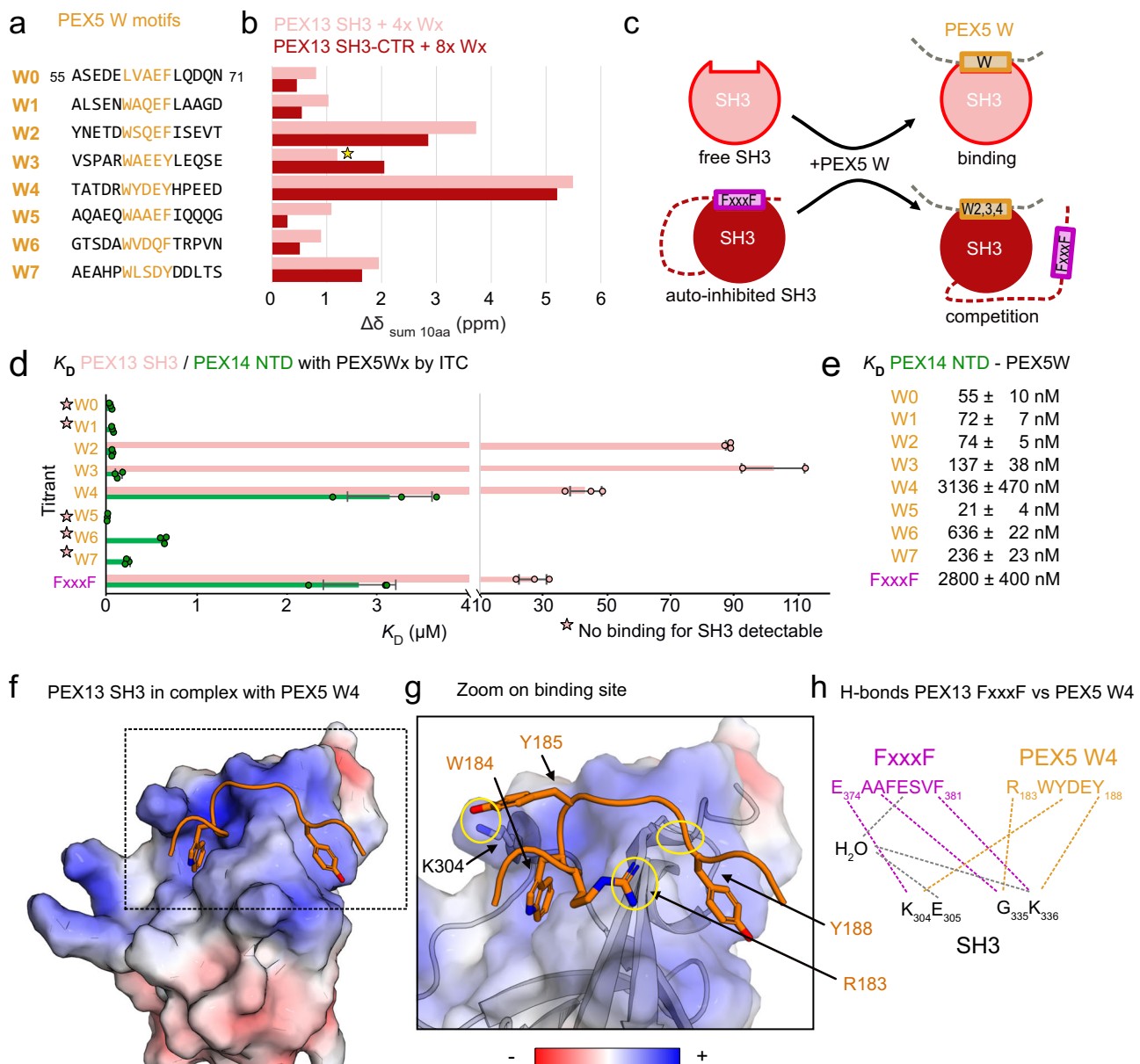

**Fig. 5 | Interaction of PEX5 (di)aromatic peptide motifs with PEX13 SH3 or SH3-CTR. a** Overview of PEX5 (di)aromatic peptide motifs. W0 was expressed as PEX5 1-76 while other W motifs were purchased as peptides as listed. **b** Induced Chemical shifts changes of PEX13 SH3 or SH3-CTR upon addition of 4x or 8x PEX5 (di) aromatic peptide motifs represented as the sum of 10 involved residues. The star indicates W3 which shows less chemical shift perturbation but extensive line-broadening. **c** Schematic representation of PEX13 SH3 / W peptide binding (top) or PEX13 SH3-CTR / W peptide competition. **d** Plot of $K_D$ values from ITC experiments of PEX13 SH3 or PEX14 NTD with PEX5 W and FxxxF peptide motifs. Average values and error bars (SD) are calculated from 3 different experiments ($n = 3$). **e** $K_D$ values from ITC experiments in numbers. **f** Electrostatic surface representation of the PEX13 SH3 GS W4 structure at 2.3 Å. **g** Zoomed view of the W binding site showing polar interactions marked with yellow circles. **h** Schematic representation of the conserved hydrogen-bond network of PEX13 SH3 / FxxxF (pink) and PEX5 W4 (orange) interaction. Additional contact sites are marked in gray. Source data are provided as a Source Data file.

displaying a GFP distribution comparable to the negative control (Fig. 8a, b). The results of all other PEX13 mutations, which enhance the intramolecular interaction of the SH3 domain (FF/W4, IA/CC), abolish FxxxF and WxxxF/Y binding (YEG) to the SH3 domain or omit interactions via the intrinsic FxxxF motif (FF/A5, YEG-FF/A5) show a comparable distribution of GFP with approximately 50% cytosolic and 50% organellar (Fig. 8a, b). Biological replicates of this experiment are shown in Supplementary Fig. 10.

These results clearly indicate that all mutations, either affecting SH3 or the FxxxF motif of PEX13, hamper the functional activity of PEX13 in peroxisomal protein import. The analysis of both cellular assays highlights the essential functional role of the PEX13 SH3 domain

and demonstrate a regulatory function of the C-terminal region including the intramolecular interaction by the FxxxF motif for PTS1 import.

**The PEX13 FxxxF motif modulates PEX5 interactions in cells**

The identified interactions between PEX13, PEX5, and PEX14 were supported in cells by immunoprecipitation experiments using the previously mentioned T-REx 293 PEX13 KO cells with re-introduced PEX13 variants. We tested PEX13 FL, the C-terminal truncation (PEX13 ΔCTR), and the two PEX13 FxxxF mutants FF/A5 and FF/W4. The immunoprecipitation experiment was set up to pull down cellular PEX5 with a PEX5 antibody. Experiments with re-introduced PEX13 FL

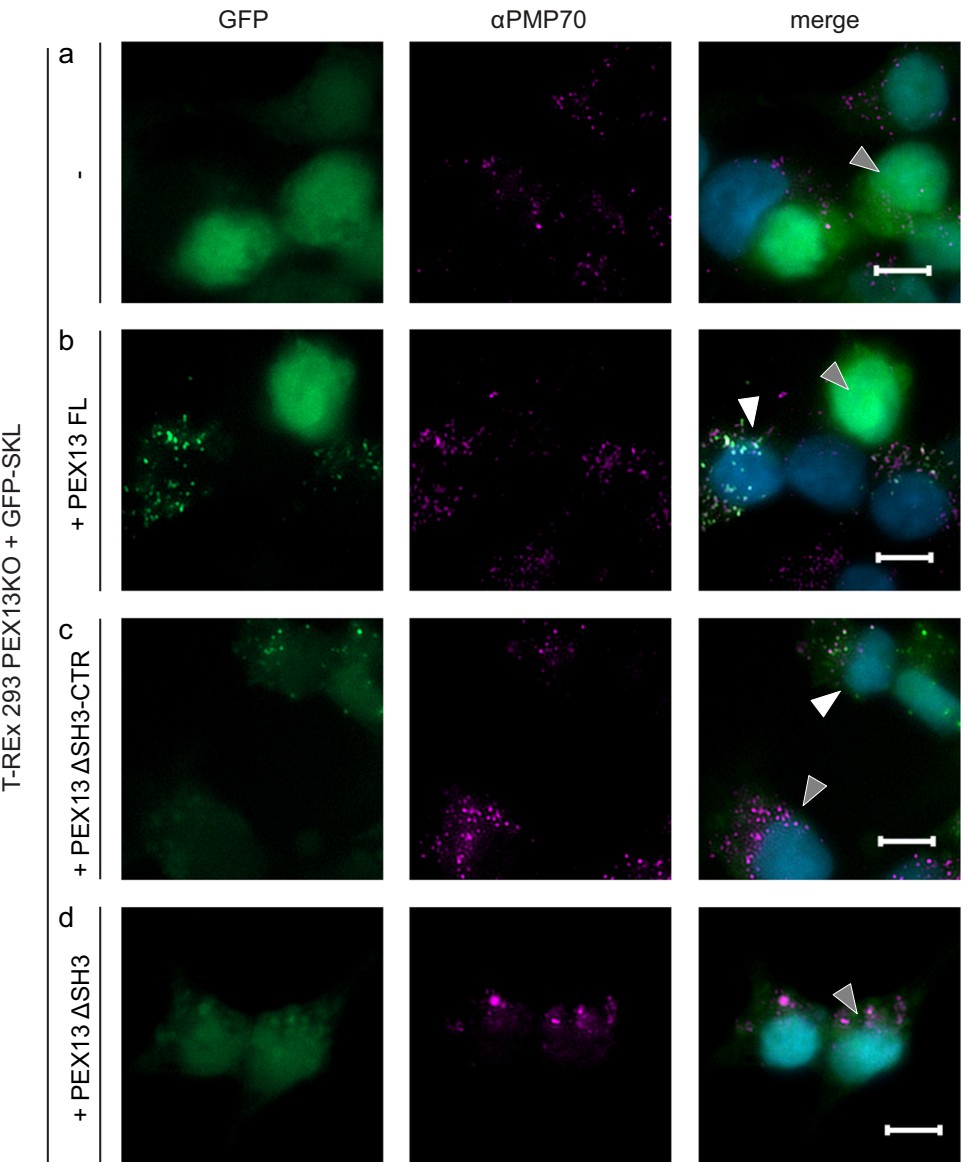

**Fig. 6 | The PEX13 SH3 domain is essential for PTS1 import.** T-REx 293 cells and PEX13-deficient (PEX13 KO) T-REx 293 cells were transfected with bicistronic expression plasmids encoding GFP-PTS1 and different PEX13 truncation and mutation variants as indicated on the left side. Rescue of the PTS1 import defect of PEX13 KO cells was monitored by fluorescence microscopy with GFP-SKL as reporter (left panels). Detection of peroxisomal membranes was achieved via PMP70 specific antibody indicated by magenta fluorescence punctate pattern (middle panels). Overlap of green and magenta dots in the merge (right panels) represent functional matrix import. Nuclei were stained with DAPI (blue, right panels). For each experiment, one cell showing no peroxisomal protein import is marked with a gray triangle and one cell showing import is marked by a white triangle, if present. Scale bar: 5 μm. In one representative analysis a total of 1216 cells were analysed. **a** Expression of GFP-SKL in PEX13 KO cells did not rescue PTS1 import indicated by a diffuse cytosolic green GFP signal (left and right panels) and served as negative control. **b** Expression of full length PEX13 led to an almost complete restored import (p < 0.0001). The introduction of a variant lacking the full C-terminal region (ΔSH3-CTR) (**c**) or the SH3 domain (ΔSH3) (**d**) did not suffi-ciently restore PTS1 import. Source data are provided as a Source Data file.

revealed visible amounts of PEX13 (Fig. 9). Upon introduction of PEX13 ΔCTR, the observed amount of PEX5-bound PEX13 was on average five times higher when compared to cells expressing PEX13 FL (Fig. 9, Supplementary Fig. 11b). An average increase of 3.5 times more PEX5-bound PEX13 compared to the PEX13 FL cell line was observed in cells expressing PEX13 FF/A5 (Fig. 9, Supplementary Fig. 11b). Replacement of the PEX13 FxxxF motif by another WxxxF motif (PEX13 FF/W4) did not affect the binding to PEX5 (Fig. 9). Noteworthy, PEX14 is present in all samples (Fig. 9). The load was controlled by the detection of GAPDH (Fig. 9, lower panel). Repeti-tions of this experiment are shown in Supplementary Fig. 11a. Overall, the data show that the absence of a functional FxxxF motif increases PEX5 binding.

## The PEX14 NTD copartitions in PEX13 condensates

Recent studies reported that PEX13 forms condensates involving its YG repeats, which are permeable for PEX5 with and without cargo[22,23]. Additionally, clusters of PEX14 and PEX13 on the peroxisomal mem-brane have been reported[23], while the orientation of PEX14 and relative orientation of the PEX14 NTD at the peroxisomal membrane is highly debated in the field. The results described above demonstrate that interactions of (di)aromatic peptide motifs in PEX5 and PEX13 with the PEX13 SH3 domain and the PEX14 NTD modulate peroxisomal import. We thus wondered whether PEX5 and PEX14 can copartition in PEX13 condensates where the identified interactions could also contribute. To this end, we tested copartitioning of PEX13 YG (His-SUMO-PEX13 40-120) condensates with PEX5 NTD (1-330) mCherry and PEX14 NTD

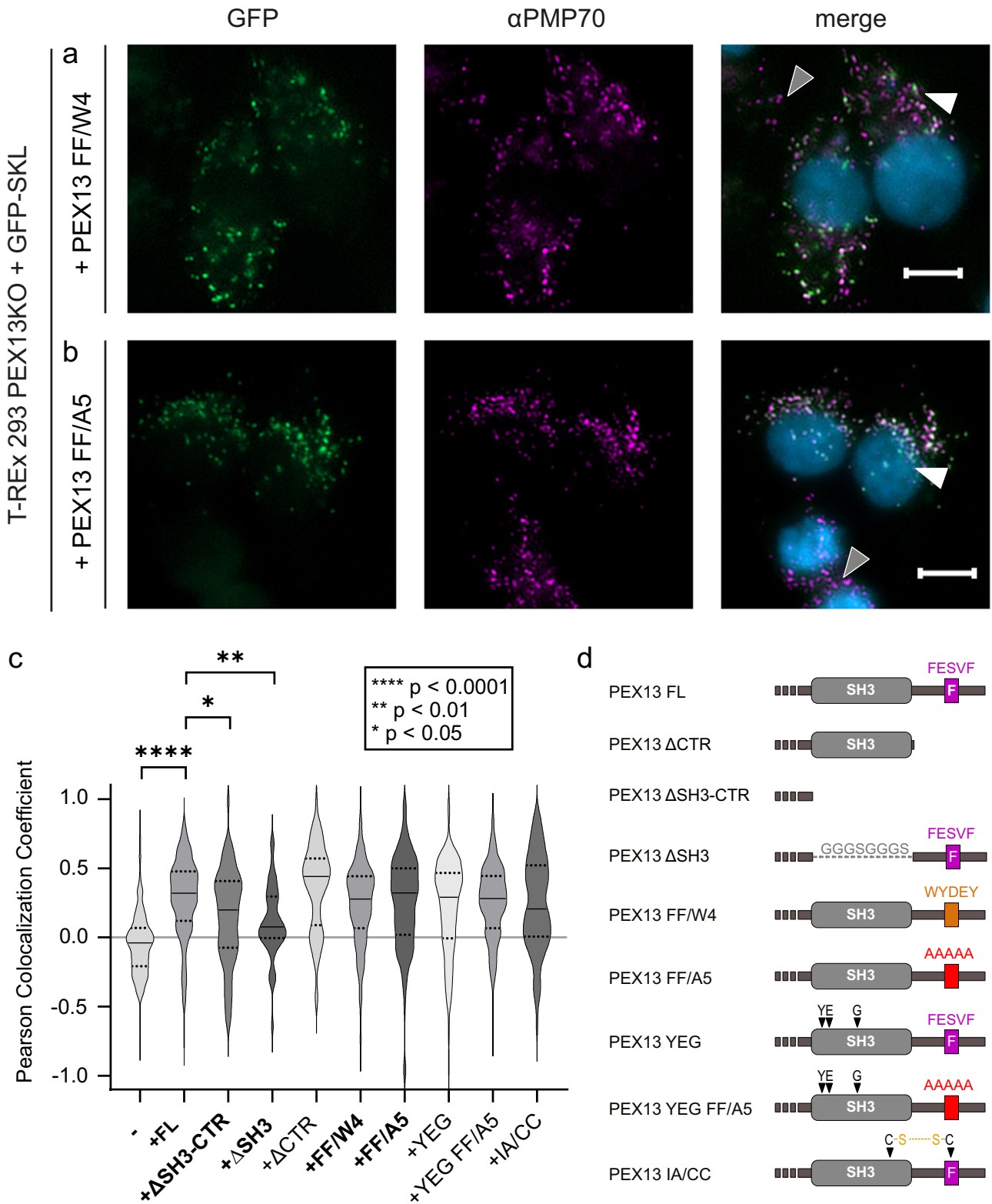

(1-104) GFP. We find that PEX13 YG forms phase-separated condensate at 100 μM but not at a concentration of 10 μM (Fig. 10a; Supplementary Fig. 12c). Under the same conditions, the negative control His-SUMO as well as PEX5 NTD or PEX14 NTD do not form condensates on their own (Fig. 10b–d). However, PEX5 NTD and PEX14 NTD copartition with PEX13 YG condensates, individually and together (Fig. 10e; Supplementary Fig. 12a, b). Strikingly, small amounts of PEX5 NTD promote condensation of PEX13 YG with PEX5 NTD at 10 μM, a concentration at

which PEX13 YG does not form condensates by itself (Supplementary Fig. 12c–e). A comparable effect is not observed with PEX14 NTD and PEX13 YG (Supplementary Fig. 12f). Taken together, these data demonstrate that PEX5 and PEX14 can copartition in PEX13 YG condensates. Furthermore, PEX14 NTD copartitions with PEX5 NTD-enhanced PEX13 YG condensates (Fig. 10e; Supplementary Fig. 12g). These results show that the critical concentration for the formation of phase-separated PEX13 YG condensates is strongly decreased by PEX5

**Fig. 7 | The PEX13 SH3-proximal FxxxF motif modulates PTS1 import.** PEX13-deficient (PEX13 KO) T-REx 293 cells were transfected with bicistronic expression plasmids encoding GFP-PTS1 and PEX13 truncation or mutation variants as indicated. Rescue of the PTS1 import defect of PEX13 KO cells was monitored by fluorescence microscopy with GFP-SKL as reporter (left panels). Detection of peroxisomal membranes was achieved via PMP70 specific antibody indicated by magenta fluorescence punctate pattern (middle panels). Overlap of green and magenta dots in the merge (right panels) represent functional matrix import. Nuclei were stained with DAPI (blue, right panels). For each experiment, a cell showing no peroxisomal protein import is marked with a gray triangle and a cell showing import is marked by a white triangle. Scale bar: 5 µm. Expression of PEX13 with enhanced (**a**) or reduced (**b**) structural autoinhibition of the SH3 domain

showed reduced PTS1 import in comparison the expression of full length PEX13. **c** The import defect from different PEX13 mutants was automated analyzed with the Pearson Colocalization Coefficient using the GFP-SKL and PMP70 signals. The Coefficient ranges from −1 (no colocalization) to +1 (perfect colocalization). For each experiment, a minimum of 45 transfected cells was analyzed. A Kruskal-Wallis test was performed for statistical analysis, and the mean rank of each cell line was compared to the mean rank of the control cell line PEX13KO complemented with PEX13FL. Correction for multiple comparisons was done with a Dunn's test. Significance is indicated as ****p < 0.0001, **p = 0.0018, *p = 0.0281. Dotted lines indicate the quartiles, while the black horizontal line indicates the median. Bold mutants are shown in main figures. **d** Schematic overview of PEX13 mutants. Source data are provided as a Source Data file.

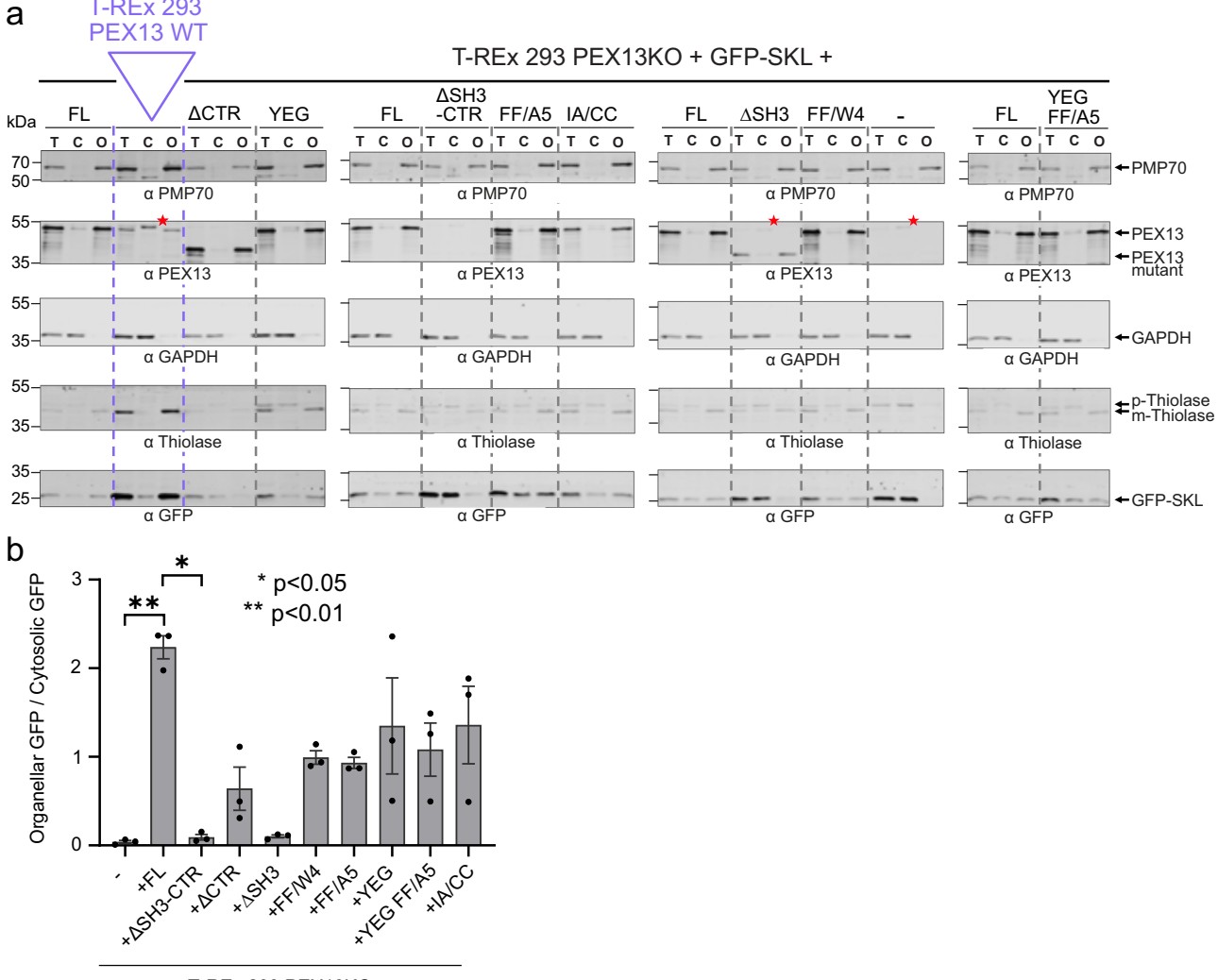

**Fig. 8 | The efficacy of PTS1- and PTS2 import is reduced in cells expressing PEX13 variants. a, b** T-REx 293 cells transfected with GFP-SKL and T-REx 293 PEX13KO cells transfected with a bicistronic vector encoding different PEX13 constructs and GFP-SKL were used for the preparation of total lysates (T) and for fractionation of the cell into cytosol and organelles (C and O, respectively). The localization of the GFP-SKL in cytosolic and organellar fractions served as readout for the PTS1 import into peroxisomes. Additionally, PTS2 import can be monitored by detecting Thiolase, which exists in a cytosolic pre-form (p-Thiolase) and in a mature, processed form (m-Thiolase) when imported into peroxisomes. **a** All immunoblots shown are from different gels of the same biological replicate. Other replicates can be found in Supplementary Fig. 10. PMP70 and GAPDH served as loading controls for the organellar and cytosolic fraction, respectively. Expression

of PEX13 variants and GFP-SKL was verified with corresponding antibodies, as indicated. PEX13 ΔSH3-CTR is not recognized by the PEX13 antibodies used as these were generated against the SH3 domain. A cross-reaction of the antibody with an unknown protein below 55 kDa is indicated by a red asterisk. **b** Total amounts of GAPDH and PMP70 in cell lysates were used to calculate the relative GFP signal in the cytosolic and organellar fractions, respectively. Average values and error bars (SEM) are calculated from 3 different experiments (*n* = 3). Black dots indicate the exact data points for each replicate. For statistical analysis a Kruskal-Wallis test was performed and the mean rank of each cell line was compared to the mean rank of the control cell line PEX13KO complemented with PEX13FL. Correction for multiple comparisons was done with a Dunn's test. Significance is indicated as **p = 0.0085 and *p = 0.0409. Source data are provided as a Source Data file.

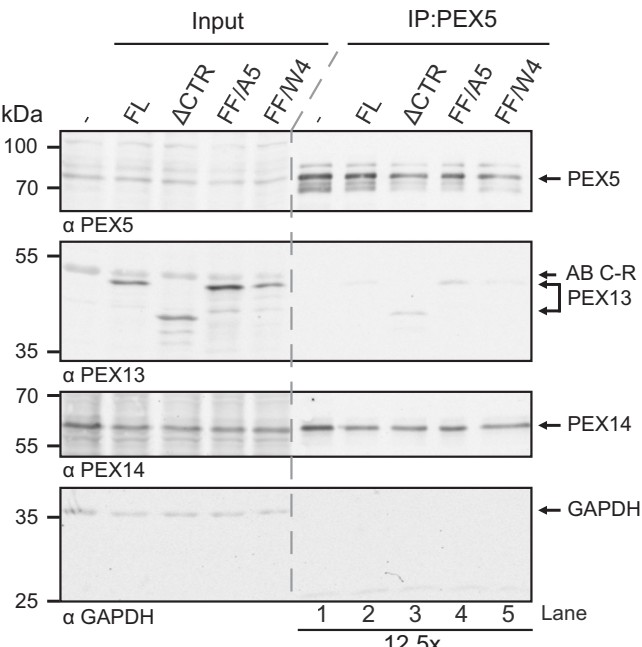

**Fig. 9 | The PEX13 FxxxF motif modulates PEX5 binding in cells.** PEX13 FL, PEX13 ΔCTR and the two mutations FxxxF to A5 and FxxxF to W4 were expressed in T-REx 293 PEX13 KO cells. The cell lysates were subjected to immunoprecipitation with PEX5 antibody and analyzed by immunoblotting. One of three independent experiments is shown, the other two experiments are shown in Supplementary Fig. 11. Source data are provided as a Source Data file.

and that the PEX14 NTD can partition with PEX13 YG or PEX13 / PEX5 condensates. This indicates that the orientation of the PEX14 relative to the peroxisomal membrane is not well defined in the presence of PEX13 condensates, which may rationalize previous apparently inconsistent models. Moreover, the network of (di)aromatic peptide motif interactions with PEX14 NTD and PEX13 SH3 may contribute to the formation of molecular interactions in biomolecular condensates and their role in peroxisomal protein translocation.

## Discussion

We present a comprehensive structural and biochemical analysis of the C-terminal region (SH3-CTR) of human PEX13 and its role in peroxisome biogenesis. We show that the PEX13 SH3 domain is essential for PTS1 import and discover that a proximal intramolecular FxxxF motif binds to the SH3 domain. We observe an interaction of the human PEX13 SH3-CTR with the PEX14 NTD mediated by the PEX13 FxxxF motif. Importantly, the SH3 domain of human PEX13 is able to recognize WxxxF/Y motifs in the PEX5 NTD, similar to yeast, demonstrating evolutionary conservation of these interactions. Strikingly, these interactions are modulated by the intramolecular interaction of the proximal FxxxF motif with the PEX13 SH3 domain. Deletion or mutation of the internal FxxxF motif or of the corresponding binding site in the SH3 domain leads to a decrease of PTS1 import efficiency. These findings demonstrate that the modulation of PEX5, PEX13, and PEX14 interactions by the SH3 FxxxF motif plays an important role in fine-tuning peroxisome biogenesis and/or matrix protein import in peroxisomes.

SH3 domains commonly recognize PxxP peptide motifs to mediate protein-protein interactions of adaptor proteins[36] in signal transduction[37]. The yeast Pex13 SH3 domain indeed recognizes a class II PxxP motif in Pex14 with the canonical PxxP binding site[25,26,38]. Strikingly, this interaction is not conserved from yeast to human as we did not observe an interaction of PxxP motifs in human PEX14 with the

PEX13 SH3 domain (Supplementary Fig. 4b). Our results imply that the interaction of the human PEX13 SH3 domain with poly-proline peptide motifs may be distinct, and involves binding to class I peptides, rationalizing why no binding with the class II PxxP motif in the N-terminal region of PEX14 is observed (Supplementary Fig. 4c, f).

The binding of human PEX13 with PEX14 does not involve an SH3 domain/PxxP interaction but is instead facilitated by binding of the PEX13 FxxxF motif to the PEX14 NTD. As expected, the recognition of the PEX13 FxxxF motif by the PEX14 NTD is structurally similar to the interaction with the (di)aromatic PEX5 peptide ligands and an FxxxF motif in the peroxisomal membrane protein transport factor PEX19[15]. This finding is in line with previous studies from Itoh and Fujiki[39], who mapped the interactions of PEX13, PEX5 and PEX19, to the PEX14 NTD (residues 21-70).

Interestingly, the yeast Pex13 SH3 domain has been shown to also recognize WxxxF/Y motifs, using a binding surface opposite to the canonical PxxP binding site. An interaction of these or similar motifs to other SH3 domains has not been reported so far. As residues in the binding site of yeast Pex13 SH3 to Pex5 WxxxF/Y are poorly conserved in human PEX13, it was unclear whether a similar interaction is also possible for human PEX13 (Supplementary Fig. 1e). Our results demonstrate that the PEX13 SH3 domain indeed binds to WxxxF/Y motifs in PEX5 and that this interaction is conserved from yeast to human. Our crystal structure of PEX13 SH3 with the W4 motif reports high-resolution details for this non-canonical binding interface of an SH3 domain fold.

A notable difference between the yeast and human PEX13 is that the interaction with WxxxF/Y motifs is modulated by intramolecular PEX13 SH3/FxxxF interaction. A previous study had mapped the PEX13 interaction with WxxxF/Y motifs from PEX5 to the N-terminal region in PEX13 rather than to the SH3 domain, while later, Krause et al. [31] proposed that the PEX13 N-terminal region mediates homo-oligomerization. These observations might be reconciled with recent reports suggesting that the N-terminal region of yeast Pex13 forms condensates, which are permeable for PEX5[22,23]. The formation of PEX13 condensates may indirectly affect the molecular interactions of the C-terminal SH3-CTR module with (di)aromatic peptide motifs.

The network of interactions identified in our study shows that PEX13, PEX5 and PEX14 interactions are modulated by binding of (di) aromatic peptide motifs with overlapping and partially competing binding sites. Thus, the formation of a ternary complex of these three peroxins as reported for yeast is not possible. This suggests an evolutionary increased complexity, where functional activity is not regulated by distinct binding motifs but by multivalency, relative affinity and avidity effects, consistent with the increasing abundance of WxxxF/Y motifs from three to eight in yeast and human PEX5, respectively[12,14,40]. The distinct binding affinities between PEX13/PEX14 (Fig. 4), PEX13/PEX5 and PEX14/PEX5 (Fig. 5) suggest a potential sequential binding model. The strong affinity of the PEX5 NTD towards PEX14 NTD may be required for initial docking of cargo-loaded PEX5 at the peroxisomal membrane. This may involve removal of co-factors that might be bound to the PEX5 NTD and thereby restrict accessibility of PEX5 W-motifs other than W0, which has been shown to be essential for targeting of PEX5 to the peroxisome[16,41,42]. The PEX14/PEX5 interaction may be later replaced by interactions involving PEX13 FxxxF/ PEX14 NTD and PEX13 SH3/PEX5 NTD. This would require a high molecular excess of PEX13 considering the lower binding affinity towards PEX5 and PEX14 and is in agreement with the observation of a high molecular mass complex, containing only PEX13 or PEX13/PEX14 from previous studies[22,23,39,43]. It was proposed that PEX14 may change interacting partners depending on the molecular size complexes of PEX14, which are disassembled in the presence of cargo-free PEX5[39]. Handing over PEX5 from PEX14 NTD to PEX13 SH3 could occur during two distinct steps of peroxisomal import; (i) as part of the docking event before cargo translocation into the peroxisomal matrix, or (ii) to

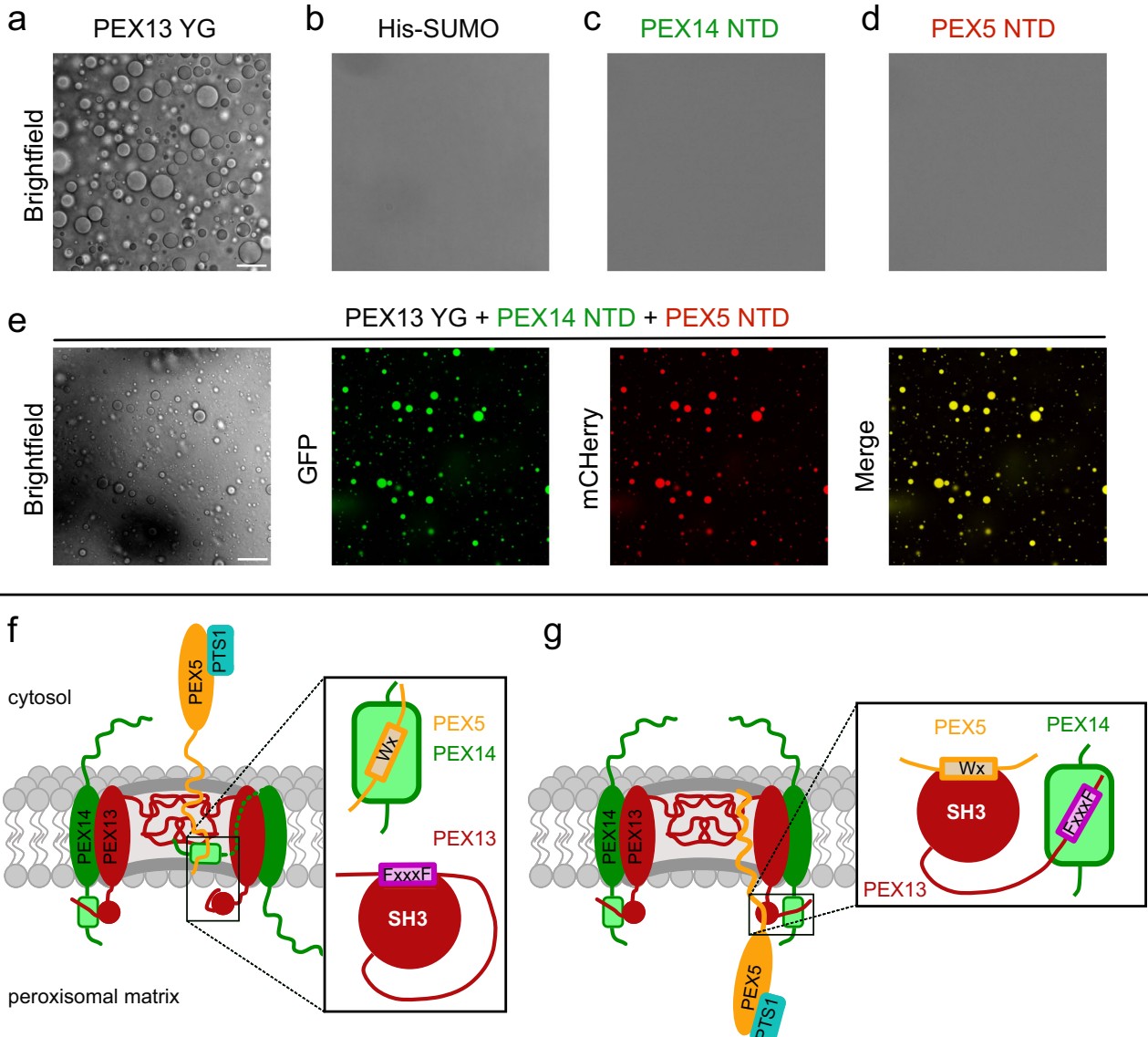

**Fig. 10 | Copartitioning of PEX14 NTD and PEX5 NTD with PEX13 YG condensates and model for the role of PEX13 in peroxisomal matrix import.** Colocalization experiments were performed in 50 mM Tris pH 7.5, 100 mM NaCl and 10% PEG8000 at RT. Scale bars indicate 25 µm. Experiments were conducted as three technical replicates. **a** At 100 µM concentration, large condensates (brightfield) are formed by PEX13 YG (His-SUMOPEX13 40–120) but not by His-SUMO (negative control) alone (**b**). Under the same conditions, PEX14 NTD (1–104) GFP (**c**) and PEX5 NTD (1–330) mCherry (**d**) do not form condensates by themselves.

**e** Condensates formed by 50 µM PEX13 YG allow simultaneous partition of PEX14 NTD (GFP) and PEX5 NTD (mCherry). **f** After a docking event, condensate formation of PEX13 PEX5 and PEX14 mediates translocation of the cargo bound PEX5 TPR domain and the PEX5 associated PEX14 NTD. **g** After translocation, a PEX13 homo-oligomer replaces the PEX5/PEX14 interaction by avidity effects handing PEX5 over to PEX13 to form a transient complex. (Boxes) Schematic representation of the interactions between PEX5, PEX14 and PEX13 in both states.

enhance cargo release inside the lumen. Involvement in docking would require the SH3 domain facing the cytosol, as it was first proposed by Gould et al. [18].

Based on recent studies contradicting orientations of SH3 relative to the peroxisomal membrane have been proposed. Experiments from rat liver indicate that the PEX13 SH3 domain faces inside the peroxisome[44], while experiments from yeast find two orientations of the Pex13 SH3 domain from which the luminal is more favored[22]. Similar to PEX13, the orientation of PEX14 is controversial and discussed in the field. Studies found PEX14 NTD to be a docking site for PEX5[15,16] and for β-tubulin[45,46], implying that the PEX14 NTD must be facing the cytosol at certain times. In contrast, proteinase K assays with peroxisomes isolated from rat liver or Xenopus eggs locate the N-terminal domain of PEX14 in the peroxisomal matrix[22,44]. Here we have assessed the potential copartitioning of PEX5 and PEX14 in PEX13

mediated biomolecular condensates. We demonstrate that the PEX14 NTD can diffuse through PEX13 YG or PEX5 NTD promoted condensates (Fig. 10e). This suggests that the PEX14 NTD and the PEX13 SH3 domain do not exhibit fixed locations, as a peroxisomal pore mediated by phase-separated condensates would connect the cytosolic side peroxisomal matrix.

Based on our data and recent reports in the field we propose a simplified mechanistic model for the role of PEX13 in peroxisomal matrix import (Fig. 10). After initial docking of PEX5/cargo complexes at the peroxisomal membrane, PEX5 copartitions with PEX13 and PEX14. This might enable the translocation of PEX14 NTD bound to the PEX5 cargo complex through the membrane (Fig. 10f). Next, the PEX5/PEX14 interaction may be replaced by the PEX13 FxxxF motif of PEX13 considering the presence of a large excess of PEX13, which forms higher homooligomers at the membrane. Concomitantly, the PEX13

FxxxF motif is released from the PEX13 SH3 domain allowing the transient binding of PEX5. PEX5, PEX13 and PEX14 now form a transient complex where PEX14 is saturated with PEX13 and PEX5 is loosely bound to PEX13 (Fig. 10g). From this intermediate, PEX5 is handed over to the export/ring finger complex consisting of PEX2, PEX10 and PEX12[47].

Our model rationalizes the utility of a network of overlapping binding sites between PEX5, PEX13, and PEX14 and resolves the apparent inconsistent observations concerning the topology of PEX14. Future studies should analyze the PEX interaction network and complexes at the membrane.

## Methods

### Molecular cloning
For recombinant expression in bacteria, the full-length DNA sequences of human PEX13 (UniProtKB no. Q92968), human PEX14 (UniProtKB no. O75381), and human PEX5 (UniProtKB no. P50542) were optimized according to the codon usage of *E. coli* and synthesized by IDT (IDT Europe GmbH, Germany). These sequences were used as templates to generate all PEX13 fragments as well as the PEX14 fragments PEX14 NTD (1-104), NTD$_{long}$ (1-113) and PEX5 W0 (1-76) in a His$_6$-SUMO-tag modified pETM13 vector (pETM13S) (Supplementary Tables 5, 6). The cloning was done using site directed ligase independent mutagenesis (SLIM)[9] in an extended version to implement inserts using the same fashion of short and tail primers. The vector backbone and inserts were amplified by polymerase chain reaction (PCR) amplification using the according short and tail primers (Supplementary Tables 5, 6) to generate overlaps with sticky ends. The backbone amplicants and inserts were mixed with a fivefold molar excess of insert and annealed during the SLIM cycle[9]. The annealed vector was directly transformed into DH10b cells for DNA amplification.

The same protocol was used to create all PEX13 constructs in the bi-cistronic mammalian expression vector pIRES2-GFP-SKL (Supplementary Table 7) and PEX5 and PEX14 GFP/mCherry constructs for condensation experiments (Supplementary Table 8).

### Protein sample preparation
PEX constructs were transformed into *Escherichia coli* BL21 (DE3) cells and expressed in LB or isotope-enriched M9 minimal medium. Uniformly $^{15}N$ or $^{15}N$, $^{13}C$ labeled proteins were expressed in H$_2$O M9 minimal medium supplemented with 50 μg/ml kanamycin, 1 g/liter [U-$^{15}N$] ammoniumchloride and 2 g/liter hydrated [U-$^{13}C$] glucose as the sole sources of nitrogen and carbon, respectively. After transformation, single colonies were picked randomly and cultured in the medium of choice overnight at 37 °C. The next day, cultures were diluted to an optical density of 600 nm (OD$_{600}$) of 0.1 and grown up to a OD$_{600}$ of 0.4-0.6. Protein expression was induced with 0.5 mM IPTG and was carried out for 4 h at 37 °C.

The cells were harvested by centrifugation at 6000 × g for 20 min at 4 °C. For protein purification the cell pellets were resuspended in lysis buffer (50 mM Tris pH 7.5, 300 mM NaCl, 20 mM imidazole) substituted with lysozyme (from chicken), DNAse and protease inhibitor mix (Serva, Heidelberg, Germany) and lysed by pulsed sonication (10 min, 40% power, large probe, Fisher Scientific model 550) followed by centrifugation at 38,000 × g for 45 min. All proteins were purified using gravity flow Ni-NTA (Qiagen, Monheim, Germany) affinity chromatography. The supernatant of the lysis was incubated with Ni-NTA beads (2 ml/1 l culture) for 20 min at 4 °C, while rotating. Subsequently to incubation, the protein-loaded beads were washed with 7 column volumes (CV) high salt buffer (50 mM Tris pH 7.5, 750 mM NaCl, 20 mM imidazole) and 10 CV wash buffer (50 mM Tris pH 7.5, 300 mM NaCl, 20 mM imidazole). The elution was performed with 3–5 CV elution buffer (50 mM Tris pH 7.5, 300 mM NaCl, 500 mM imidazole). Dialysis and SUMO cleavage was executed over night at 4 °C in 20 mM Tris pH 7.5, 150 mM NaCl. Further purification was done with a reverse

Ni-NTA column where the flow through containing the cleaved protein of interest was collected and concentrated for size exclusion chromatography using a Superdex S75, 16/600 (Cytiva, Marlborough, US). The size exclusion chromatography as last step of the purification was performed directly in NMR buffer.

### Peptides
All 15-mer peptides used in this study were purchased from PSL (Peptide Specialty Laboratories GmbH, Heidelberg, Germany). Peptides delivered in TFA salt were dissolved in H$_2$O and the pH adjusted to 7.5 using 1 M NaOH. If used for ITC, the peptides were dialyzed against ITC buffer (20 mM Tris pH 7.5, 50 mM NaCl).

| | |
|---|---|
| PEX5 W1 | ALSENWAQEFLAAGD |
| PEX5 W2 | YNETDWSQEFISEVT |
| PEX5 W3 | VSPARWAEEYLEQSE |
| PEX5 W4 | TATDRWYDEYHPEED |
| PEX5 W5 | AQAEQWAAEFIQQQG |
| PEX5 W6 | GTSDAWVDQFTRPVN |
| PEX5 W7 | AEAHPWLSDYDDLTS |

### NMR spectroscopy
NMR data were collected on Bruker Avance III spectrometers operating at 500, 600, 800, 900 or 950 MHz, equipped with cryogenic probes. The sequential assignment of backbone resonances for PEX13 SH3-CTR was performed based on heteronuclear experiments such as $^1H$-$^{15}N$-HSQC, HNCA, HN(CO)CA, CBCA(CO)NH, HNCACB, HNCO, HN(CA)CO, HN(CA)NNH and H(NCA)NN[48,49]. {$^1H$}-$^{15}N$ heteronuclear NOE (hetNOE)[50] experiments were performed using the pulse sequence hsqcnoef3gpsi (Bruker, Avance version 12.01.11) with a 4.5 s interscan delay. NOE values are given simply by the ratio of the peak heights in the experiment with and without proton saturation (hetNOE = I$_{sat}$/I$_0$)[51]. $^{15}N$ HSQC-based $T_1$ and $T_2$ experiments used sequences developed from Farrow, et al. [50] with water-control during the relaxation period in the $T_1$ sequence using a cosine-modulated IBURP-2 pulse[52] and modifications in the $T_2$ sequences based on[53]. For both $T_1$ and $T_2$ experiments 8 time points with delays of 80, 160, 240, 320, 400, 64, 800, 1000 ms ($T_1$) and 14.4, 28.8, 43.2, 57.6, 72.0, 86.4, 100.8, 115.2 ms ($T_2$) were measured respectively. NMR-Spectra were processed using Topspin (Bruker Biospin, Rheinstetten, Germany) or NMRPipe[54] and analyzed using CcpNMR Analysis 2.4.2[55].

All NMR experiments were performed at 298°K. PEX13 and PEX14 spectra were recorded in 20 mM Tris pH 7.5, 50 mM NaCl and 50 mM NaP pH 6.5 and 100 mM NaCl respectively.

For all titration experiments a reference protein concentration 100 μM was used. For protein/protein titrations, every titration point was prepared as individual sample to avoid dilution effects. Protein/peptide titrations such as titration of PEX5 (di)aromatic peptide motifs with high concentrated peptides (10–15mM) were performed in a single NMR tube. Ligands were added with increasing concentrations up to an excess of eightfold. The chemical shift perturbation ($\Delta\delta_{avg}$) was calculated by using formula $\Delta\delta_{avg} = [(\Delta\delta_H) + (\Delta\delta_N * 0.159)^2]^{0.5}$. Chemical shift perturbations in Fig. 5b are illustrated with the sum of 10 residues which are affected upon (di)aromatic peptide motif binding. NMR chemical shift assignments are available at the BMRB, accession code: 51336.

### Isothermal titration calorimetry (ITC)
Isothermal titration calorimetry (ITC) measurements were performed as triplicates at 25 °C using a MicroCal PEAQ-ITC (Malvern Instruments Ltd. U.K) calorimeter. Buffer conditions were 20 mM Tris pH 7.5, 50 mM NaCl. For all titrations a titrant dilution control experiment was performed and subtracted before the data were fitted to a one-site binding model using the Malvern Analysis software.

PEX13 SH3 at a concentration of 35-48 μM was titrated with purchased 15mer peptides containing (di)aromatic penta peptides from

PEX5 or PEX5 1-76 (W0) at concentration of 0.7-0.6 mM and PEX13 350-403 (FxxxF) at a concentration of 0.9-1 mM. PEX14 (1-104) at a concentration of 20-60 μM was titrated with PEX5 W peptides or PEX5 W0 at concentration of 0.4 to 0.6 mM, PEX13 FxxxF at a concentration of 0.46 mM and with PEX13 SH3-CTR at a concentration of 1 mM. The concentration of PEX14 was corrected with the fit, since it cannot be accurate measured at 280 nm owing to the extinction coefficient of only 1490.

## X-ray crystallography

All crystals were grown using the vapor diffusion sitting drop method in 96 well plates. Therefore, the proteins were purified in 5 mM Tris pH 7.5, 50 mM NaCl and later screened using commercial crystallization sets (200 nl drops, 1:1 ratio). All proteins were crystallized at a concentration of 20 mg/ml. The PEX 13 SH3 domain alone was crystallized in 0.01 M Zinc chloride, 0.1 M sodium acetate pH 5 and 20 % (w/v) PEG6000. PEX13 SH3 2GSc FxxxF chimera was crystallized in 0.2 M sodium chloride, 0.1 M Bis-Tris pH 6.5 and 25% PEG3350. PEX13 SH3 GSc PEX5 W4 chimera was crystallized in 0.2 M sodium sulfate, 0.1 M Bis Tris propane pH 6.5 and 20% (w/v) PEG3350. All crystals were cryoprotected using 25% ethylene glycol prior to flash freezing. Data of apo PEX13 SH3 and PEX13 SH3-2GSc-FxxxF (PEX13$_{1-346}$-GGGGSGGGGSDEQEAAFESVFVE) chimera were collected at SLS beamline X06DA (Paul-Scherer-Institute, Villigen, Switzerland) while data of PEX13 SH3-GSc-W4 (PEX13$_{1-346}$-GGGGSTATDRWYDEYHPEE) chimera was collected at beamline P11 at PETRA III (EMBL, Hamburg, DE).

Collected data were processed using CCP4i2 software suite[56,57]. XDS was used for data indexing and reduction, aimless for data scaling, MOLREP[58] and PHASER[59] for molecular replacement, COOT[60] for model building and REFMAC5 for refinement[61]. Refined structures were uploaded to wwPDBdeposition using pdb_extract[62].

## Size exclusion chromatography−static light scattering (SEC-SLS)

SLS on PEX13 SH3-CTR was done using an OmniSEC Resolve and Reveal device (Malvern Panalytics. Malvern, Uk) equipped with a Superdex 75 increase 10/300 GL column (Cytiva). First 70 μl 2 mg/ml BSA standard (column calibration) and then 70 μl of 2 mg/ml PEX13-SH3-CTR in NMR/ITC buffer (20 mM Tris pH 7.5, 50 mM NaCl) was passed though the column with a constant flow of 0.3 ml/min. The concentration was monitored via absorption at 280 nm and the refractive index. The molecular weight was calculated with the RALS signal using Omnisec software (version 11.01, Malvern Panalytics, Malvern, Uk).

## Computational modeling

The structure of PI3K SH3 domain in complex with a PxxP ligand (PDB ID: 3I5R) was identified as similar to PEX13 SH3 domain by sequence search. Both domains were aligned and the PxxP ligand was copied to the PEX13 SH3 structure and subsequently mutated to PEX13 PxxP (TRVPPPIL) using Maestro (Schrodinger suite). The energy of the complex was then minimized using OPLS2005 force field and freely rotation of all residues in a radius of 4 Å.

## Multiple sequence alignments

**Multiple sequence alignment of PEX13 SH3 domains.** First, a RCSB PDB databank search for structures of human SH3 domains was performed. Eight human SH3 domains from different proteins as well as yeast Pex13p SH3 domain were selected (Supplementary Table 9). Then the according sequences including the ± 10 flanking amino acids were selected and with the sequence from the human PEX13 SH3 aligned using clustalΩ (https://toolkit.tuebingen.mpg.de/tools/clustalo) and visualized using Jalview (version 2.11.2.0).

**Multiple sequence alignment of PEX13 with mammalian sequences.** Mammalian sequences, which are similar to PEX13 were found using PSI-BLAST from NCBI selecting the non-redundant protein sequences database with the organism restricted to mammals. The maximal target sequences were adjusted to 250 and the run started with preselected standards. From the hits, 186 sequences with a query coverage of minimum 90% were selected and aligned using clustalΩ (https://toolkit.tuebingen.mpg.de/tools/clustalo) before being visualized using ConSurf web server[63,64] (https://consurf.tau.ac.il/).

## Cell culture

T-REx™293 (Invitrogen, USA) and T-REx 293 PEX13KO cells[35] were grown in Dulbecco's Modified Eagle's Medium (DMEM) containing 25 mM D-Glucose and 4 mM L-Glutamine (Gibco, UK), supplemented with 10 % fetal calf serum, 100,000 U/l penicillin and 100 mg/l streptomycin at 37 °C and 5% $CO_2$. For cell transfections, X-tremeGENE HP DNA Transfection Reagent (Roche, Germany) was used according to the manufacturer's instructions.

## Fluorescence microscopy

To perform immunofluorescence microscopy, cells were seeded on coverslips to appropriate density and fixed with 3% formaldehyde/D'PBS for 20 min. After membrane permeabilization with 1% Triton X-100/D'PBS for 5 min a blocking step was performed by incubation in 1% BSA/D'PBS for 1 h. Cells were then incubated with the primary antibody targeting PMP70 (rabbit, 1:500 in D'PBS, Invitrogen PA1-650) for 30 min. Afterwards, the secondary antibody goat-α-rabbit IgG Alexa Fluor 594 (Invitrogen) was added for 10 min under light protection. Cells were mounted on glass slides with Mowiol 4-88 (Calbiochem, USA) supplemented with DAPI. Imaging was performed using the Axio Imager M2 microscope (Zeiss, Germany).

Quantification of peroxisomal protein import was done using the colocalization tool of the Zen blue edition software (Zeiss). Threshold values for the signal intensities of GFP and PMP70 were determined by comparison with stained, non-transfected cells and applied to all analyzed images. The Pearson Colocalization Coefficient was determined for at least 45 individual cells per condition. Statistical analysis was done using Prism (GraphPad) and applying a Kruskal-Wallis test. Correction for multiple comparisons was performed using Dunn's test.

## Cell lysis and fractionation

For lysis and fractionation, cells were seeded into 60 mm cell culture plates and grown to ~70% density after transfection. Cells were then washed once with Hanks Balanced Salt Solution (HBSS, Sigma Aldrich, Germany), trypsinized and collected in HBSS supplemented with calcium chloride and magnesium sulfate (Sigma Aldrich). Half of the volume was used for total lysis while the other half was used for fractionation. In both cases cells were harvested by centrifugation at 200 × g and 4 °C for 5 min. Cells submitted to total lysis were resuspended in RIPA buffer (150 mM NaCl, 50 mM Tris, 1% (v/v) IGEPAL, 0.5% (w/v) sodium deoxycholate, 0.1% (w/v) SDS, pH 8.0, supplemented with 1x protease inhibitor cocktail (Roche)). Cells submitted to fractionation were resuspended in the same volume of 0.005% Digitonin/D'PBS, supplemented with 1x protease inhibitor cocktail (Roche). Both samples were incubated on ice for 30 min before centrifugation at 6800 × g and 4 °C for 5 min. SDS samples were taken from both supernatants (total and cytosolic fraction respectively). The remaining pellets of the fractionation samples were resuspended in RIPA buffer with protease inhibitor cocktail before incubation on ice for 30 min and subsequent centrifugation at 6800 × g and 4 °C for 5 min. An SDS sample of the supernatant was prepared for analysis of the organellar fraction. Samples were analyzed by SDS PAGE and immunoblotting was performed onto nitrocellulose membrane with 0.45 μm pore size (GE Healthcare, Germany). Antibodies used for immunoblotting

recognized PMP70 (Invitrogen PA1-650), Thiolase (Sigma Aldrich V2139), GAPDH (Proteintech 60004-1-Ig), GFP (Santa Cruz sc-9996) and PEX13 (Proteintech 26649-1-AP) and were diluted in 1x PBS, supplemented with 0.1% TritonX-100. Protein bands were visualized with corresponding secondary antibodies, either IRDye® 680 or IRDye® 800CW (LI-COR Biotechnology, Germany). Immunoblot signal intensities were quantified using ImageJ software. GAPDH and PMP70 were used as specific loading controls for cytosolic and organellar samples, respectively. Statistical analysis was done with the Prism software (GraphPad) using a Kruskal-Wallis test and correcting for multiple comparisons with Dunn's test.

## Co-immunoprecipitation

To study interactions between PEX5 and PEX13 dynabeads were coupled with a mouse PEX5 antibody[65], using the dynabeads™ Antibody Coupling Kit (Invitrogen, USA) according to the manufacturer's instructions.

Cells were seeded on 10 cm dishes and transfected with different PEX13 truncations or mutations. 48 h after transfection cells were incubated in IP lysis buffer (25 mM Tris/HCl pH 7.4, 150 mM NaCl, 1% NP-40, 5% glycerol, cOmplete™ EDTA-free Protease Inhibitor Cocktail (Roche), 25 μg/ml DNase) for 15 min on ice. After removal of cell debris by centrifugation (16,000 × g, 5 min, 4 °C) equal amounts of lysates were incubated with dynabeads on a rotating disk for 1 h at 4 °C. Next the dynabeads were washed three times with lysis buffer and bound proteins were eluted with 0.1 M Glycin pH 2.8. Samples were collected and analyzed by SDS-PAGE and immunoblotting using the following antibodies: rabbit aPEX5 (1:5000), rabbit aPEX13 (1:1000, Proteintech, Germany)[66], chicken aPEX14 (1:1000, Ruhr-University Bochum), mouse aGAPDH (1:7500, Proteintech, Germany). Band intensities on immunoblots were quantified using densitometry (ImageJ, NIH).

## Fluorescence microscopy for condensation experiments

Condensation experiments were carried out on a Leica TCS SP8 confocal microscope using a 63x water immersion objective (1.2 NA). Samples (V = 30 μL) in 50 mM Tris pH 7.5, 100 mM NaCl and 10% PEG8000 were prepared by mixing in a 0.5 ml reaction tube and transferred to micro-well plates (Ibidi, μ-Slide Angiogenesis Glass Bottom). Samples containing GFP-labeled proteins were excited using the 488 nm laser line and imaged at 495-550 nm, while the 552 nm laser line and an emission window between 580 and 650 nm was used for mCherry-labeled proteins. Additionally, a bright field image was acquired on all experiments. When more than one fluorophore was present in the sample, both channels were recorded simultaneously. To rule out bleed-through, single-labeled controls were carried out maintaining acquisition parameters constant.

## Reporting summary

Further information on research design is available in the Nature Portfolio Reporting Summary linked to this article.

## Data availability

The data generated or analyzed during this study are included in this article and its supplementary file. Source data are provided with this paper. NMR chemical shift assignments of PEX13 SH3-CTR are available at the BMRB ID: 51336. Crystallographic data of the PEX13 SH3 domain, PEX13 SH3 domain complexed with the FxxxF motif and the PEX13 SH3 domain complexed with the PEX5-W4 motif are available at the PDB, with accession codes: 7Z0I (PEX13 SH3-FxxxF), 7Z0J (PEX13 SH3), 7Z0K (PEX13 SH3-PEX5-W4 complex). Source data are provided with this paper.

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

## Acknowledgements

We thank Sam Asami, Mark Bostock, Gerd Gemmecker for support with NMR experiments. Access to NMR measurements at the Bavarian NMR center and the X-ray Crystallography Platform at Helmholtz Zentrum München is acknowledged. We acknowledge access to the Paul Scherrer Institut, Villigen, Switzerland for provision of synchrotron radiation beamtime at beamline X06DA of the SLS and thank Florian Dworkowski for assistance. Other synchrotron data were collected at beamline P11 operated by EMBL Hamburg at the PETRA III storage ring (DESY, Hamburg, Germany). We would like to thank Johanna Hakanpää for the assistance in using the beamline. This work was supported by the Deutsche Forschungsgemeinschaft (FOR1905, project number 219314758, SA823/11-2, TP05) to M.S. and R.E. (ER178/6-2, ER178/7-2, ER178/17-1), and by the German Federal Ministry of Research and Education (BMBF) program "Targetvalidierung für die pharmazeutische Wirkstoffentwicklung" (GFTARV38) and BMBF VIP+ (03VP05531). F.D. acknowledges support by an EMBO Long-term Fellowship (ALTF 243-2018).

## Author contributions

M.S., S.G. and R.E. designed the study. S.G. performed protein expression, biophysical and NMR experiments. S.G. and F.D. analyzed NMR experiments. S.G. and K.Z. performed X-ray crystallography, S.G., K.Z. and G.P. analyzed crystallographic data. S.G., J.S. and J.B. performed and analyzed LLPS experiments. R.P., J.O. and W.S. planned and performed pulldown and complementation assays. S.G., M.S. and R.E. wrote the manuscript, all authors commented and approved the manuscript.

## Funding

## Competing interests

The authors declare no competing interests.
