## [Peer Review File · Nature Communications]

REVIEWER COMMENTS

Reviewer #1 (Remarks to the Author):

The authors present a very nice and clean structural study on the interaction of PAX 5, 13 and 14 that are crucial players in peroxisomal import. The authors describe an interaction of the Pex13 SH3 domain with a diaromatic repeat located in its own C-terminus by X-ray crystallography. To this end, special constructs are made that are called chimeras, cutting off most of the C-terminal floppiness. Furthermore, chemical shift titrations with subsections of the three proteins are presented and their interplay is discussed with respect to functional consequences, cross checked by LSM experiments. The presented results deserve publication in Nature Communication.

The writing is somewhat unclear here and there. It is not stated in the right way what the advance of this paper is, in comparison to the 2005 paper by the Erdmann group or even in comparison to earlier ones by others. Of course, I understand that it is in the structural work and especially characterization of the interplay of interactions but even this needs to be said more carefully. As examples, neither the end of the introduction nor the first titles in the results sections are clear on this aspect. The first sentence in the last section of the introduction says 'In this study we present the first structural characterization of the Pex13 SH3 domain and....' which is not really intended by the authors, I guess. There are 5 Pex13 SH3 domain structures in the PDB, some with a bound peptide. Maybe replace 'and' with 'with connected C-terminal....' or describe the construct(s) more comprehensively. The first title in the result section says 'The Pex13 C-terminal region harbors an SH3 domain followed by a FxxxF motif' which is something that should be said in the introduction since it is known since a long time. The content of this section would be better covered by another title. And there are more examples. Other than that, the paper presents the interaction study clearly and is important for understanding peroxisome function.

Minor points:

The sequences of the crystallized constructs are described in a convolute way. They should be stated explicitly.

I am not sure that this FxxxF repeat should be called motif, since it appears only in this Pex family of proteins. 'Repeat' would be appropriate, since a 'motif' would require more than 30 references appearing when typing this into the Web of Science, in my humble opinion.

Furthermore, in Figs. 1C and 3C I do not see any asterisks that could indicate prolines or missing assignments; however, the Figure legends mention them. How much is assigned?

Reviewer #2 (Remarks to the Author):

In this manuscript, the authors use several techniques to explore the role of a C-terminal FxxxF domain in the essential peroxisome biogenesis protein PEX13. They demonstrate that this PEX13 FxxxF motif

binds to the PEX13 SH3 domain (Fig. 2), that this binding can be competed off by PEX5 N-terminal W motifs (Fig. 5), that the PEX13 FxxxF motif binds to the NTD of PEX14 (Fig. 3, 4), that the PEX13 SH3 domain promotes matrix protein import (Fig. 6C, D), and that the region C-terminal of the SH3 domain impairs matrix protein import if mutations are made that in the FxxxF domain that increase or decrease PEX5 binding (Fig. 6E, 6F, 7).

In summary, the strength of the paper is in the structural analysis and individual binding assays, but the importance of these interactions and how they together modulate import of proteins into the peroxisome is not yet clear.

Major comments:

1. I am not convinced that the authors have demonstrated “autoinhibition” as claimed in the title and elsewhere in the manuscript. The authors show that the C-terminal PEX13 FxxxF domain binds to the PEX13 SH3 domain. But the term “inhibition” implies that this binding inhibits PEX13 activity. The activity of PEX13 is to promote matrix protein import, and the only experiment testing the role of the FxxxF domain in matrix protein import in Figure 6. Deleting the entire C-terminal domain decreases import slightly, suggesting that something about the C-terminal domain facilitates rather than inhibits import. Moreover, specifically changing the FxxxF domain’s affinity for the PEX13 SH3 domain or for PEX5 do not alter import in ways that are interpretable by an inhibition model. Thus, there is no support in this figure for “autoinhibition” of PEX13 by the FxxxF domain that is referred to in the manuscript title and throughout the manuscript and would be more accurate to simply describe the binding that has been observed.

2. Independent of these interpretation problems, there is no statistical support indicating which import numbers are significantly different in Figure 6. It appears from the methods that the authors combined the data from three biological replicates (p. 15) rather than treating them separately to allow statistical analysis.

3. The data of Figure 6 includes cells with partial import and complete import both scored as import positive. It would be more transparent to categorize and report cells in three bins: full import, partial import, or complete import. Also, it would be preferable not to adjust the wild-type PEX13 import to 100% when only 81% of cells showed import (p. 15). Simply reporting the import that was observed (with statistics) would be easier for the reader to understand.

4. In Figure 6, I did not find where the authors indicate that the different constructs produce similar levels of protein in the cells. If these levels are shown in Figure 7, then it should be explicitly stated.

5. A recent Science paper (Gao et al., 2022, ref 49) has substantially advanced our understanding of matrix protein import, including the conserved and central role that PEX13 plays in this process. Although this Gao et al. paper primarily centered on the yeast Pex13 protein, and although the FxxxF domain of interest to the authors of the current manuscript is not present in the yeast protein, it is disappointing that the authors do not account for these recent findings in their models for matrix protein import or when interpreting their data. For example,

- The Gao paper findings should be discussed and referenced in the introduction when introducing matrix protein import and when introducing previous findings germane to PEX13 function. For example, the statement “Pex13 might not be directly involved in the translocation process” (p. 3) seems out of date.
- In addition, the PEX13 domain structure in Figure 1B could be updated to denote the N-terminal YG domain and the long central amphipathic helix that were functionally highlighted in the Gao paper and implicated in transport of PEX5-Cargo complexes across the peroxisome membrane.
- The Gao paper provides multiple lines of convincing evidence that PEX13 is present in both orientations in the peroxisome membrane and plays a central role in transporting cargo into the matrix. This dual topology has implications for several of the authors’ experiments and their interpretation. The simplest model for matrix protein import that is consistent with all the data may no longer be the PEX14-centric model shown in Figure 1A.
- The Gao paper (along with other papers; e.g. Skowyra and Rapoport, ref 51) provides convincing evidence on the topology of PEX14 (N in, C out) in the peroxisome membrane (from various organisms) that is not consistent with the model proposed in Figure 8 with PEX14 mysteriously flipping its orientation in the membrane with each import cycle. Although this prior work does not eliminate the possibility of this movement, it is not the simplest explanation of the data given the new data.

The manuscript is a bit hard to follow at times and could be clarified by addressing the following minor points:

6. p. 3 - The statement in the introduction that Pex17 “seems to be dispensable in humans” would be clearer as “is not found in humans”. (Dispensable implies present but not needed.)

7. p. 3 – The statement about the essentiality of Pex17 for peroxisomal import in yeast should be accompanied by a reference.

8. p. 4 – The first section heading of the results section is not descriptive of the results in this section. Perhaps “The PEX13 C-terminal region harbors two structured elements – an SH3 domain followed by a FxxxF motif.”

9. p. 4 – the statement that the FxxxF motif is “highly conserved across mammals” would be more complete with an addendum adding some specificity, e.g., “but is not found in yeast, plants, or lower vertebrates” or whatever (depending on where it is found).

10. p. 4 - The heading “The PEX13 SH3 domain is autoinhibited by the C-terminal FxxxF motif” implies that inhibition is being measured. This section seems only to monitor binding of the FxxxF motif to the SH3 domain, and so the section heading is not reflective of the findings. Perhaps “The PEX13 FxxxF motif binds to the PEX13 SH3 domain”?

11. The change suggested above would also make the heading parallel with the following section heading, which might then be “the PEX13 FxxxF motif also binds the PEX14 NTD.”

12. p. 5 – The phrase “a PEX13 FxxxF construct (...residues 350-403)” is confusing. Is this the same as “the isolated FxxxF peptide” in the next paragraph? If so, please unify the nomenclature. If not, please indicate which amino acids comprise “the isolated FxxxF peptide.”

13. p. 7 – On page 7, the authors begin referring to the C-terminal FxxxF motif of PEX13 as the “internal FxxxF motif”. It would be less confusing to use “C-terminal FxxxF” or “PEX13 FxxxF” rather than “internal”, as “internal” calls to mind the middle domain of a protein. The heading on p. 7 would be clearer as “PEX5 WxxxF/Y motifs compete with the PEX13 FxxxF motif for binding to the PEX13 SH3 domain” rather than “PEX5 WxxxF/Y motifs compete with the internal FxxxF motif on PEX13 SH3.”

14. p. 7 – Fig. 4E is mentioned on this page, but there is no panel E in Figure 4.

15. p. 7 – “encoding for...” can be replaced by “encoding...” or “coding for...”

16. p. 8 – consider changing “which was supposed to...” to “which we expected to...”

17. p. 16 – It seems that the “Funding” and “Acknowledgments” headings are switched.

18. Reference 48 is a BioRxiv manuscript, which is not clear from the citation.

19. “Noteworthy” is an adjective and cannot be used to start a sentence in the context of “Noteworthy, a previous study...” The authors might consider “Notably,” instead in these instances.

20. Figure 6 - In the schematics that accompany Figure 6B, it would be informative to have the amino acids “FESVF” above the “F” box as the “AAAAA” and “WYDEY” are above the corresponding boxes in panels E and F.

21. Figure 7 – It would be clearer to combine Figure 7A and B into one set of schematics with the sequences in Figure 7B replaced by the schematics in Figure 6E and F. Also the amino acids (FESVF) could be added above the “F” box in Figure 7A as suggested for Figure 6B to unify the schematics.

Reviewer #3 (Remarks to the Author):

In this paper, Gaussmann et al. report on the autoinhibition of PEX13 peroxisomal import, mediated by interactions between the SH3 domain of PEX13 and a newly identified FxxxF motif in PEX13. The authors further investigate interactions between PEX13 and PEX14 mediated by the FxxxF motif. The authors use a combination of NMR, X-ray crystallography, molecular biology, ITC, fluorescence microscopy, and immunoprecipitation studies to probe the mechanistic role of PEX13 in peroxisomal matrix import and further propose a model involving interactions with PEX5 and PEX14.

Sections of the paper – the author’s scientific approach and results are somewhat interesting, but overall the narrative and flow of this manuscript are plagued by a number of issues that need to be corrected. My expertise primarily limits my experimental feedback to the structural and biophysical sections of this manuscript. In general, the crystal structures PDB 7Z01, 7Z0J, and 7Z0K have been processed and refined to the standards of the structural biology community. Their NMR correlation data is sufficient to support the PEX13 binding model proposed by the authors. I commend the authors’ efforts to correlate their NMR and X-ray crystallographic data, and use of ITC to measure KD’s between motifs they observed interactions in.

At this time, I do not recommend acceptance of this manuscript to Nat. Comms. and instead suggest the authors submit to a more specialized journal. I do not believe the subject matter of this manuscript to be of sufficient interest or impact to the readership of Nat. Comms. The scientific approach – though detailed, valid, and presented with the appropriate controls is not novel. The results and “mechanistic model” proposed by the authors ostensibly represent an incremental step in our knowledge of peroxins, rather than a fundamental shift in thinking toward addressing a knowledge gap.

In this paper, Gaussmann et al. report on the autoinhibition of PEX13 peroxisomal import, mediated by interactions between the SH3 domain of PEX13 and a newly identified FxxxF motif in PEX13. The authors further investigate interactions between PEX13 and PEX14 mediated by the FxxxF motif. The authors use a combination of NMR, X-ray crystallography, molecular biology, ITC, fluorescence microscopy, and immunoprecipitation studies to probe the mechanistic role of PEX13 in peroxisomal matrix import and further propose a model involving interactions with PEX5 and PEX14.

Sections of the paper – the author’s scientific approach and results are somewhat interesting, but overall the narrative and flow of this manuscript are plagued by a number of issues that need to be corrected. My expertise primarily limits my experimental feedback to the structural and biophysical sections of this manuscript. In general, the crystal structures PDB 7Z01, 7Z0J, and 7Z0K have been processed and refined to the standards of the structural biology community. Their NMR correlation data is sufficient to support the PEX13 binding model proposed by the authors. I commend the authors’ efforts to correlate their NMR and X-ray crystallographic data, and use of ITC to measure KD’s between motifs they observed interactions in.

At this time, I do not recommend acceptance of this manuscript to Nat. Comms. and instead suggest the authors submit to a more specialized journal. I do not believe the subject matter of this manuscript to be of sufficient interest or impact to the readership of Nat. Comms. The scientific approach – though detailed, valid, and presented with the appropriate controls is not novel. The results and “mechanistic model” proposed by the authors ostensibly represent an incremental step in our knowledge of peroxins, rather than a fundamental shift in thinking toward addressing a knowledge gap.

Please find below some textual comments that the authors may find helpful.

Abstract:

An interaction **between** of PEX13 and PEX14 involves the PEX13 FxxxF motif.

Surprisingly, the canonical PxxP binding surface on the SH3 domain **of PEX13** does not recognize PxxP motifs in PEX14, distinct from the yeast ortholog.

Introduction:

The first callout for Fig 1A should occur much earlier. Perhaps at the end of this sentence in paragraph 1.

Peroxisomal cargo proteins with their destination in the peroxisomal matrix possess conserved peroxisomal targeting signals, at their C-terminus (PTS1) or N-terminus (PTS2) (Fig. 1A)^{13,14}.

I believe that this is the first call out to the SH3 domain in the main text, post-abstract, the authors should clarify that they are referring to a protein domain, as they have with their NTD abbreviation.

Many aspects of peroxisome biogenesis have been studied in yeast, where Pex5/cargo docking is

mediated by a membrane-associated complex ~~consistent of the~~ **that comprises** the Pex13 SH3 domain, and the NTDs of the Pex14 NTD and Pex5 NTD, **essential for the import of cargoes labeled with either both PTS1 or and PTS2 import** 17,25-27,30

Paragraph 1 of the introduction requires a paragraph break. I suggest starting a new paragraph at the following sentence. I further suggest that the author provide additional references at the end of this sentence to support their statement.

The general mechanisms of peroxisomal biogenesis and matrix protein import are evolutionary conserved.

The interactions within the docking complex are mediated **by interactions between** (di)aromatic penta-peptide motifs (“WxxxY”) **found within Pex5**, ~~of Pex5~~ with the N-terminal domain of PEX14, and a poly-proline (PxxP) motif of Pex14 that binds to the Pex13 SH3 domain 31-33.

Results:

Paragraph 1 is very well written, but requires a paragraph break to allow readers the opportunity to better parse the author’s NMR data. I suggest starting a new paragraph at the following sentence.

We then compared NMR correlation spectra of the PEX13 SH3-CTR and SH3 domain.

Paragraph 2 requires a paragraph break. I suggest starting a new paragraph with the following sentence.

Interestingly, nine out of the eleven Arg and Lys residues are located at the FxxxF binding surface, causing a highly positive charge, which is favorable for binding negative charged peptides such as the C-terminal FxxxF motif (**Fig. 2C**).

Readers would benefit from having the β 1 and β 5 strands labeled in Fig. 2A, in addition to Supplementary Fig. 2B.

Analysis of the structure of the SH3 domain shows a network of polar contacts between the N- and C-terminal regions **that stabilizing interactions between** the β 1 and β 5 ~~interaction~~ strands (**Supplementary Fig. 2B**), **which is** consistent with the extended domain boundaries observed by NMR. The structure of ~~the this~~ complex **further reveals** ~~shows that~~ interactions **between** ~~of~~ the α -helical FxxxF motif and SH3 domain **are** mediated by hydrophobic contacts ~~of~~ **between Phe# and Phe##** ~~the two phenylalanines~~, which clamp around **the β 1 and β 5 strands** (**Fig. 2A, C**). ~~and~~ Polar **hydrogen-bonding** interactions **between the backbone and sidechains** **provide further stability** (**Fig. 2B**). ~~involving sidechain and backbone contacts.~~

The crystal structure was confirmed in solution by NMR titration experiments of the isolated SH3 domain with a FxxxF peptide, **residues 350–403**, showing strong chemical shift perturbations in the binding site expected from the crystal structure, where the spectrum at saturated binding is very similar to the native SH3-CTR (**Supplementary Fig. 2D, E**).

Point-by-point response

Reviewer #1 (Remarks to the Author):

The authors present a very nice and clean structural study on the interaction of PAX 5, 13 and 14 that are crucial players in peroxisomal import. The authors describe an interaction of the Pex13 SH3 domain with a diaromatic repeat located in its own C-terminus by X-ray crystallography. To this end, special constructs are made that are called chimeras, cutting off most of the C-terminal floppiness. Furthermore, chemical shift titrations with subsections of the three proteins are presented and their interplay is discussed with respect to functional consequences, cross checked by LSM experiments. The presented results deserve publication in Nature Communication.

The writing is somewhat unclear here and there. It is not stated in the right way what the advance of this paper is, in comparison to the 2005 paper by the Erdmann group or even in comparison to earlier ones by others. Of course, I understand that it is in the structural work and especially characterization of the interplay of interactions but even this needs to be said more carefully. As examples, neither the end of the introduction nor the first titles in the results sections are clear on this aspect. The first sentence in the last section of the introduction says 'In this study we present the first structural characterization of the Pex13 SH3 domain and.....' which is not really intended by the authors, I guess. There are 5 Pex13 SH3 domain structures in the PDB, some with a bound peptide. Maybe replace 'and' with 'with connected C-terminal....' or describe the construct(s) more comprehensively. The first title in the result section says 'The Pex13 C-terminal region harbors an SH3 domain followed by a FxxxF motif' which is something that should be said in the introduction since it is known since a long time. The content of this section would be better covered by another title. And there are more examples. Other than that, the paper presents the interaction study clearly and is important for understanding peroxisome function.

Thank you for these comments and suggestions. We have addressed these with the following changes:

We make it now clearer that we focus on human PEX13, and stress that our results present (i) the first high resolution structure of the human SH3 domain and (ii), more importantly, the first high resolution structures of an SH3 domain in complex with an FxxxF or WxxxF motif, bound at a non-canonical surface on the "backside" of the SH3 fold, i.e. opposite to the canonical PxxP binding site.

The FxxxF motif is now already mentioned in the introduction.

Minor points:

The sequences of the crystallized constructs are described in a convolute way. They should be stated explicitly.

We have checked the description and make sure to provide a clear description of the crystallization construct. The corresponding sequences are reported in the methods section.

I am not sure that this FxxxF repeat should be called motif, since it appears only in this Pex family of proteins. 'Repeat' would be appropriate, since a 'motif' would require more than 30 references appearing when typing this into the Web of Science, in my humble opinion.

Please note that such amino acid sequence motifs are found in the whole genome. Moreover, the sequence is listed in motif databases such as ELM (<http://elm.eu.org/>).

Please note that the sequences are also referred to as "motif" by others (see Gao et al., Science 2022).

We would therefore prefer to the FxxxF sequence as a motif.

Furthermore, in Figs. 1C and 3C I do not see any asterisks that could indicate prolines or missing assignments; however, the Figure legends mention them. How much is assigned?

Thank you for this suggestion. We have changed the size of the asterisks to be better visible.

Reviewer #2 (Remarks to the Author):

In this manuscript, the authors use several techniques to explore the role of a C-terminal FxxxF domain in the essential peroxisome biogenesis protein PEX13. They demonstrate that this PEX13 FxxxF motif binds to the PEX13 SH3 domain (Fig. 2), that this binding can be competed off by PEX5 N-terminal W motifs (Fig. 5), that the PEX13 FxxxF motif binds to the NTD of PEX14 (Fig. 3, 4), that the PEX13 SH3 domain promotes matrix protein import (Fig. 6C, D), and that the region C-terminal of the SH3 domain impairs matrix protein import if mutations are made that in the FxxxF domain that increase or decrease PEX5 binding (Fig. 6E, 6F, 7).

In summary, the strength of the paper is in the structural analysis and individual binding assays, but the importance of these interactions and how they together modulate import of proteins into the peroxisome is not yet clear.

Thank you for the appreciation of our structural biochemical data. We provide new experimental data and have revised the text and discussion to better highlight the functional roles of our findings.

Major comments:

1. I am not convinced that the authors have demonstrated “autoinhibition” as claimed in the title and elsewhere in the manuscript. The authors show that the C-terminal PEX13 FxxxF domain binds to the PEX13 SH3 domain. But the term “inhibition” implies that this binding inhibits PEX13 activity. The activity of PEX13 is to promote matrix protein import, and the only experiment testing the role of the FxxxF domain in matrix protein import in Figure 6. Deleting the entire C-terminal domain decreases import slightly, suggesting that something about the C-terminal domain facilitates rather than inhibits import. Moreover, specifically changing the FxxxF domain’s affinity for the PEX13 SH3 domain or for PEX5 do not alter import in ways that are interpretable by an inhibition model. Thus, there is no support in this figure for “autoinhibition” of PEX13 by the FxxxF domain that is referred to in the manuscript title and throughout the manuscript and would be more accurate to simply describe the binding that has been observed.

Thank you for the suggestion to clarify this aspect. In response to this comment, we have changed the title and revised the text. The structure of the SH3-CTR shows an intramolecular interaction, which we consider a structural inhibition of the SH3 domain. We agree that it may be misleading if this would automatically imply a functional autoinhibition. The new title reads: *‘Modulation of peroxisomal import by the PEX13 SH3 domain and a proximal FxxxF binding motif’*. Throughout the manuscript, we now refer to the structural features as intramolecular interaction. Our functional studies show that the intramolecular SH3/FxxxF interaction and the interplay of (di)aromatic peptide motifs binding to PEX13 SH3 and PEX14 NTD modulates peroxisomal import. The text has been revised accordingly.

2. Independent of these interpretation problems, there is no statistical support indicating which import numbers are significantly different in Figure 6. It appears from the methods that the authors combined the data from three biological replicates (p. 15) rather than treating them separately to allow statistical analysis.

To address this point, we repeated all experiments and did evaluate the import efficiency of cells expressing different PEX13 variants with the Pearson Colocalization Coefficient (PCC) using automated correlation of PMP70 (Alexa Fluor 594) and GFP fluorescence signals. The coefficient ranges from -1 (no colocalization) to +1 (perfect colocalization). The data are shown in the new **Figure 7**. As outlined in the legend, for each experiment, a minimum of 45 transfected cells was analyzed. A Kruskal-Wallis test was performed for statistical analysis, and the mean rank of each cell line was compared to the mean rank of the control cell line PEX13KO complemented with PEX13FL. Correction for multiple comparisons was done with a Dunn’s test. Significance is indicated as **** $p < 0.0001$, ** $p < 0.01$, * $p < 0.05$.

3. The data of Figure 6 includes cells with partial import and complete import both scored as import positive. It would be more transparent to categorize and report cells in three bins: full import, partial import, or complete import. Also, it would be preferable not to adjust the wild-type PEX13 import to 100% when only 81% of cells showed import (p. 15). Simply reporting the import that was observed (with statistics) would be easier for the reader to understand.

To address this point, we searched for a more sensitive method that would allow a quantification of the import in cells. To this end, we applied digitonin fractionation of cells with subsequent analysis via immunoblotting to dissect the localization of GFP. The analysis is shown in the new **Figure 8**. As indicated in the legend, total amounts of GAPDH and PMP70 in cell lysates were used to calculate the relative GFP signal in the cytosolic and organellar fractions, respectively. Immunoblot signal intensities were quantified using the ImageJ software. The quantification and standard deviation shown in Figure 8B is calculated from three independent biological replicates.

4. In Figure 6, I did not find where the authors indicate that the different constructs produce similar levels of protein in the cells. If these levels are shown in Figure 7, then it should be explicitly stated.

To address this point, the expression levels were analyzed by immunoblotting as shown in Figure 8, which shows that the different constructs indeed produce similar levels of protein in the cells.

5. A recent Science paper (Gao et al., 2022, ref 49) has substantially advanced our understanding of matrix protein import, including the conserved and central role that PEX13 plays in this process. Although this Gao et al. paper primarily centered on the yeast Pex13 protein, and although the FxxxF domain of interest to the authors of the current manuscript is not present in the yeast protein, it is disappointing that the authors do not account for these recent findings in their models for matrix protein import or when interpreting their data. For example,

- The Gao paper findings should be discussed and referenced in the introduction when introducing matrix protein import and when introducing previous findings germane to PEX13 function. For example, the statement “Pex13 might not be directly involved in the translocation process” (p. 3) seems out of date.

Thank you for these suggestions. Please note that our manuscript was submitted just at the time of the publication of the Gao paper. We agree and have revised the text of the introduction accordingly, and also discuss the recent findings from Gao *et al* and Ravindra *et al* under consideration of the data of this manuscript. We would like to note that the study of Gao *et al.*, 2022 focusses on the yeast machinery.

In our manuscript, we now point out that many, but not all mechanistic aspects of peroxisomal import are conserved from yeast to human. We also note that the minimal pore in yeast required for functional PTS1 consists of scPex14/scPex17 and scPex5 (Meinecke et al., 2010, Nat Cell Biol), while there is no human Pex17 homolog known.

For this reason, we highlight that our study focusses on the human peroxisome. Moreover, to address this comment, we added new experiments demonstrating that human PEX13 forms biomolecular condensates with co-partitioning PEX5 and PEX14, depending on different regions involved. These experiments are shown in the newly added **Figure 10**. The phase-separation depends on intrinsically disordered regions, such as the PEX5 NTD and PEX13 YG motifs. We suggest that the (di)aromatic motif interactions with PEX14 NTD and PEX13 SH3 may play a role for condensate formation. Our experiments with the human proteins demonstrate that the phase-separated condensate formation for peroxisomal translocation likely extends to the human pore.

- In addition, the PEX13 domain structure in Figure 1B could be updated to denote the N-terminal YG domain and the long central amphipathic helix that were functionally highlighted in the Gao paper and implicated in transport of PEX5-Cargo complexes across the peroxisome membrane.

We agree and have updated Figure 1A and 1B, which now considers the condensate formation involving the PEX13 YG domain and the roles of the PEX5 NTD for cargo translocation.

The presence of an amphipathic helix is likely conserved but the amino acid sequences of these regions are not strongly conserved between yeast and human. In fact, previous studies from Barros-Barbosa et al., 2019 proposed at least 3 transmembrane spans using a proteinase K protection assay. However, we agree with the model of an amphipathic helix, which would also protect this region from proteinase K digest and thus adjusted Fig. 1B.

- The Gao paper provides multiple lines of convincing evidence that PEX13 is present in both orientations in the peroxisome membrane and plays a central role in transporting cargo into the matrix. This dual topology has implications for several of the authors' experiments and their interpretation. The simplest model for matrix protein import that is consistent with all the data may no longer be the PEX14-centric model shown in Figure 1A.

We agree that PEX13 should be considered as a central component of the translocation machinery, and have updated Figure 1 accordingly. However, for the sake of simplicity, we do not show domains in Figure 1 but consider that the PEX5 NTD can co-partition with PEX13 condensates and thereby support translocation of PEX5/cargo complexes, as suggested.

Nevertheless, it is important to note that details of the yeast and human translocation machinery may be distinct. Barros-Barbosa et al (2019) analyzed the topology of PEX13 and PEX14 in detail and came to the conclusion that the C-terminal SH3-domain of human PEX13 is facing the lumen, while the C-terminal region of human PEX14 is facing the cytosol. Thus, the specific details of the pore and translocation may be distinct between yeast and human. Accordingly, we refer to the published topologies of the human proteins in the model shown in Figure 10.

- The Gao paper (along with other papers; e.g. Skowrya and Rapoport, ref 51) provides convincing evidence on the topology of PEX14 (N in, C out) in the peroxisome membrane (from various organisms) that is not consistent with the model proposed in Figure 8 with PEX14 mysteriously flipping its orientation in the membrane with each import cycle. Although this prior work does not eliminate the possibility of this movement, it is not the simplest explanation of the data given the new data.

We respectfully do not agree with this statement as different topologies have been reported and well supported by experimental data (Bharti *et al.*, 2011; Neufeld *et al.*, 2009; Neuhaus *et al.*, 2014; Shimizu *et al.*, 1999). Furthermore, the human PEX14 NTD has been shown to be the docking site for β -tubulin (Reuter *et al.*, 2021). We therefore conclude that the PEX14 NTD must be facing the cytosol at certain times. In this context, we now provide experimental data to show that the PEX14 NTD can co-partition with PEX13-mediated condensates, which may provide a possible explanation for the apparently conflicting reports on different PEX14 topologies. In these condensates which would span the peroxisomal membrane cytosolic and lumen regions would be similarly accessible, and distinct interactions that vary through the import cycle may occur facing different sites of the membrane.

The manuscript is a bit hard to follow at times and could be clarified by addressing the following minor points:

6. p. 3 - The statement in the introduction that Pex17 “seems to be dispensable in humans” would be clearer as “is not found in humans”. (Dispensable implies present but not needed.)

Changed accordingly.

7. p. 3 – The statement about the essentiality of Pex17 for peroxisomal import in yeast should be accompanied by a reference

We have added a statement on this and introduced a reference in the introduction.

8. p. 4 – The first section heading of the results section is not descriptive of the results in this section. Perhaps “The PEX13 C-terminal region harbors two structured elements – an SH3 domain followed by a FxxxF motif.

Changed accordingly.

9. p. 4 – the statement that the FxxxF motif is “highly conserved across mammals” would be more complete with an addendum adding some specificity, e.g., “but is not found in yeast, plants, or lower vertebrates” or whatever (depending on where it is found)

We agree and have clarified this statement at page 4, which now reads: ‘...., which is highly conserved across mammals (Supplementary Fig. 1C), but not found in yeast or invertebrates’.

10. p. 4 - The heading “The PEX13 SH3 domain is autoinhibited by the C-terminal FxxxF motif” implies that inhibition is being measured. This section seems only to monitor binding of the FxxxF motif to the SH3 domain, and so the section heading is not reflective of the findings. Perhaps “The PEX13 FxxxF motif binds to the PEX13 SH3 domain”?

Changed accordingly. Our new mutational analysis also shows the intra- vs intermolecular affinities of the FxxxF motif.

11. The change suggested above would also make the heading parallel with the following section heading, which might then be “the PEX13 FxxxF motif also binds the PEX14 NTD.”

Changed accordingly.

12. p. 5 – The phrase “a PEX13 FxxxF construct (...residues 350-403)” is confusing. Is this the same as “the isolated FxxxF peptide” in the next paragraph? If so, please unify the nomenclature. If not, please indicate which amino acids comprise “the isolated FxxxF peptide.”

Changed accordingly.

13. p. 7 – On page 7, the authors begin referring to the C-terminal FxxxF motif of PEX13 as the “internal FxxxF motif”. It would be less confusing to use “C-terminal FxxxF” or “PEX13 FxxxF” rather than “internal”, as “internal” calls to mind the middle domain of a protein. The heading on p. 7 would be clearer as “PEX5 WxxxF/Y motifs compete with the PEX13 FxxxF motif for binding to the PEX13 SH3 domain” rather than “PEX5 WxxxF/Y motifs compete with the internal FxxxF motif on PEX13 SH3.”

Changed accordingly.

14. p. 7 – Fig. 4E is mentioned on this page, but there is no panel E in Figure 4.

Changed accordingly.

15. p. 7 – “encoding for...” can be replaced by “encoding...” or “coding for...”

Changed accordingly.

16. p. 8 – consider changing “which was supposed to...” to “which we expected to...”

Changed accordingly.

17. p. 16 – It seems that the “Funding” and “Acknowledgments” headings are switched.

Changed accordingly.

18. Reference 48 is a BioRxiv manuscript, which is not clear from the citation.

Changed accordingly.

19. “Noteworthy” is an adjective and cannot be used to start a sentence in the context of “Noteworthy, a previous study...” The authors might consider “Notably,” instead in these instances.

Changed accordingly.

20. Figure 6 - In the schematics that accompany Figure 6B, it would be informative to have the amino acids "FESVF" above the "F" box as the "AAAAA" and "WYDEY" are above the corresponding boxes in panels E and F.

Changed accordingly.

21. Figure 7 – It would be clearer to combine Figure 7A and B into one set of schematics with the sequences in Figure 7B replaced by the schematics in Figure 6E and F. Also the amino acids (FESVF) could be added above the "F" box in Figure 7A as suggested for Figure 6B to unify the schematics.

Figure 7 was replaced by a new figure. All schematics of the constructs contain the amino acid sequence now.

Reviewer #3 (Remarks to the Author):

In this paper, Gaussmann et al. report on the autoinhibition of PEX13 peroxisomal import, mediated by interactions between the SH3 domain of PEX13 and a newly identified FxxxF motif in PEX13. The authors further investigate interactions between PEX13 and PEX14 mediated by the FxxxF motif. The authors use a combination of NMR, X-ray crystallography, molecular biology, ITC, fluorescence microscopy, and immunoprecipitation studies to probe the mechanistic role of PEX13 in peroxisomal matrix import and further propose a model involving interactions with PEX5 and PEX14.

Sections of the paper – the author's scientific approach and results are somewhat interesting, but overall the narrative and flow of this manuscript are plagued by a number of issues that need to be corrected.

We have substantially revised the manuscript text, introduction, presentation and discussion of our data, and provide new experiments to support our conclusions.

My expertise primarily limits my experimental feedback to the structural and biophysical sections of this manuscript. In general, the crystal structures PDB 7Z01, 7Z0J, and 7Z0K have been processed and refined to the standards of the structural biology community. Their NMR correlation data is sufficient to support the PEX13 binding model proposed by the authors. I commend the authors' efforts to correlate their NMR and X-ray crystallographic data, and use of ITC to measure KD's between motifs they observed interactions in.

Thank you.

At this time, I do not recommend acceptance of this manuscript to Nat. Comms. and instead suggest the authors submit to a more specialized journal. I do not believe the subject matter of this manuscript to be of sufficient interest or impact to the readership of Nat. Comms. The scientific approach – though detailed, valid, and presented with the appropriate controls is not novel. The results and "mechanistic model" proposed by the authors ostensibly represent an incremental step in our knowledge of peroxins, rather than a fundamental shift in thinking toward addressing a knowledge gap.

No specific points are mentioned. However, we would like to refer to reviewer 1 and 2 who acknowledge, that our manuscript provides novel insight into the role of PEX13 in peroxisome biogenesis and protein translocation.

With the new experimental data included in the revised manuscript and the revised text we believe that we present a strong manuscript with interesting novel findings on the role of PEX13.

REVIEWERS' COMMENTS

Reviewer #2 (Remarks to the Author):

The authors have been very responsive to the previous reviewer comments and the revision has added new supporting data and addressed many of the previous limitations. The new model in Figure 10 and the new schematics throughout will help the reader understand the complex findings that are being communicated. The text does a better job of incorporating recent results from other labs and indicating which findings are controversial.

Major comments:

1. The quantitative results of Figure 8 displayed in Figure 8B would be clearer as a single bar for each genotype representing the fraction organellar GFP. This change would allow a statistical comparison of the complementation ability of the various PEX13 derivatives. Because the slight differences among these constructs in restoring PTS1 import are not statistically significant in Figure 7C, it is important to have some statistical support for the differences observed in Figure 8B, as these differences are a major claim of the manuscript.

2. The results of Figure 9 (PEX13-PEX5 interactions; lines 310-324) are described in quantitative terms, but no quantification is provided. If the authors wish to conclude that “the absence of a function FxxxF motif increases PEX5 binding ... (and)... the presence of a more potent WxxxF motif at the C-terminus hampers the PEX5-PEX13 interaction” (lines 322-324), then some quantification of the co-IP results should be provided.

Minor points:

Line 113 – Please check if “Supplementary Fig. 1C” be “Supplementary Fig. 1E.”

Line 225 – Please check whether figure call-outs are correct.

Line 375 – Consider changing “Not unexpected” to “As expected” (or “Unsurprisingly”) to avoid the double-negative.

Line 427 – Consider changing “On the opposite,” to “In contrast,”

Line 859 – Consider changing “the peroxins” to “the human peroxins” in the legend to Figure 1 to remind the reader that the summary does not include data from the yeast proteins.

Figure 4D – It would be useful to remind the reader that red is PEX13 and green is PEX14 in the model (as is done in Figures 10F and G).

Point-by-point response

Reviewer #2

The authors have been very responsive to the previous reviewer comments and the revision has added new supporting data and addressed many of the previous limitations. The new model in Figure 10 and the new schematics throughout will help the reader understand the complex findings that are being communicated. The text does a better job of incorporating recent results from other labs and indicating which findings are controversial.

Thank you.

Major comments:

1. *The quantitative results of Figure 8 displayed in Figure 8B would be clearer as a single bar for each genotype representing the fraction organellar GFP. This change would allow a statistical comparison of the complementation ability of the various PEX13 derivatives. Because the slight differences among these constructs in restoring PTS1 import are not statistically significant in Figure 7C, it is important to have some statistical support for the differences observed in Figure 8B, as these differences are a major claim of the manuscript.*

Thank you for the comment. The analysis in Fig. 8b is now presented as bar chart with statistical analysis of the GFP localization. The text has been revised accordingly.

2. *The results of Figure 9 (PEX13-PEX5 interactions; lines 310-324) are described in quantitative terms, but no quantification is provided. If the authors wish to conclude that “the absence of a function FxxxF motif increases PEX5 binding ... (and)... the presence of a more potent WxxxF motif at the C-terminus hampers the PEX5-PEX13 interaction” (lines 322-324), then some quantification of the co-IP results should be provided.*

The quantification of the results in Fig. 9 is now presented in Supplementary Fig. 11b. The text was revised accordingly.

Minor points:

Line 113 – Please check if “Supplementary Fig. 1C” be “Supplementary Fig. 1E.”

That is correct, thank you

Line 225 – Please check whether figure call-outs are correct.

Thank you, we checked and corrected all figure call-outs

Line 375 – Consider changing “Not unexpected” to “As expected” (or “Unsurprisingly”) to avoid the double-negative.

changed to “As expected”

Line 427 – Consider changing “On the opposite,” to “In contrast,”

changed to “In contrast”

Line 859 – Consider changing “the peroxins” to “the human peroxins” in the legend to Figure 1 to remind the reader that the summary does not include data from the yeast proteins.

Figure 4D – It would be useful to remind the reader that red is PEX13 and green is PEX14 in the model (as is done in Figures 10F and G).

Thank you, colored labeling was added to Fig. 4d